# Reverse Engineering Self-Supervised Learning

**Ido Ben-Shaul**
Department of Applied Mathematics
Tel-Aviv University & eBay Research
`ido.benshaul@gmail.com`

**Ravid Shwartz-Ziv**[*]
New York University
`ravid.shwartz.ziv@nyu.edu`

**Tomer Galanti**[*]
Massachusetts Institute of Technology
`galanti@mit.edu`

**Shai Dekel**
Department of Applied Mathematics
Tel-Aviv University
`shaidekel6@gmail.com`

**Yann LeCun**
New York University & Meta AI, FAIR
`yann@cs.nyu.edu`

## Abstract

Self-supervised learning (SSL) is a powerful tool in machine learning, but understanding the learned representations and their underlying mechanisms remains a challenge. This paper presents an in-depth empirical analysis of SSL-trained representations, encompassing diverse models, architectures, and hyperparameters. Our study reveals an intriguing aspect of the SSL training process: it inherently facilitates the clustering of samples with respect to semantic labels, which is surprisingly driven by the SSL objective's regularization term. This clustering process not only enhances downstream classification but also compresses the data information. Furthermore, we establish that SSL-trained representations align more closely with semantic classes rather than random classes. Remarkably, we show that learned representations align with semantic classes across various hierarchical levels, and this alignment increases during training and when moving deeper into the network. Our findings provide valuable insights into SSL's representation learning mechanisms and their impact on performance across different sets of classes.

## 1 Introduction

Self-supervised learning (SSL) [12, 5] has made significant progress over the last few years, almost reaching the performance of supervised baselines on many downstream tasks [42, 4, 31, 30, 47, 34, 15, 37, 66, 16]. However, understanding the learned representations and their underlying mechanisms remains a persistent challenge due to the complexity of the models and the lack of labeled training data. Moreover, the pretext tasks used in self-supervision frequently lack direct relevance to the specific downstream tasks, further complicating the interpretation of the learned representations.

In supervised classification, however, the structure of the learned representations is often very simple. For instance, a recent line of work [50, 36] has identified a universal phenomenon called neural collapse, which occurs at the terminal stages of training. Neural collapse reveals several properties of the top-layer embeddings, such as mapping samples from the same class to the same point, maximizing the margin between the classes, and certain dualities between activations and the top-layer weight matrix. Later studies [10, 28, 26, 56, 62] characterized related properties at intermediate layers of the network.

---

[*]Equal contribution

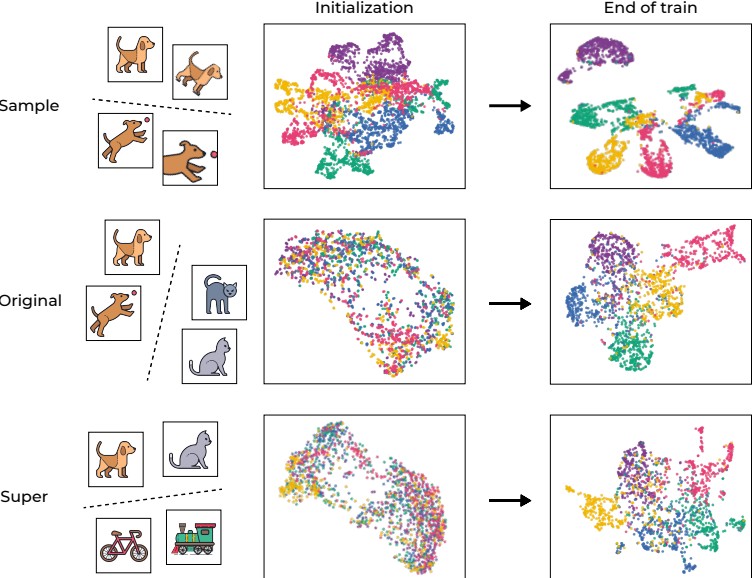

Figure 1: **SSL training induced semantic clustering.** UMAP visualization of the SSL representations before and after training in different hierarchies. **(top)** Augmentations of five different samples, each sample colored distinctly. **(middle)** Samples from five different classes within the standard CIFAR-100 dataset. **(bottom)** Samples from five different superclasses within the dataset.

Compared to traditional classification tasks that aim to accurately categorize samples into specific classes, modern SSL algorithms typically minimize loss functions that combine two essential components: one for clustering augmented samples (invariance constraint) and another for preventing representation collapse (regularization constraint). For example, contrastive learning methods [14, 48, 63, 46] train representations to be indistinguishable for different augmentations of the same sample and at the same time to distinguish augmentations of different samples. On the other hand, non-contrastive approaches [6, 66, 34] use regularizers to avoid representation collapse.

**Contributions.** In this paper, we provide an in-depth analysis of representation learning with SSL through a series of carefully designed experiments. Our investigation sheds light on the clustering process that occurs during training. Specifically, our findings reveal that augmented samples exhibit highly clustered behavior, forming centroids around the mean embedding of augmented samples sharing the same original image. More surprisingly, we observe that clustering also occurs with respect to semantic labels, despite the absence of explicit information about the target task. This demonstrates the capability of SSL to capture and group samples based on semantic similarities.

Our main contributions are summarized as follows:

- **Clustering:** We investigate the clustering properties of SSL-trained representations. In Figure 2, we show that akin to supervised classification [50], SSL-trained representation functions exhibit a centroid-like geometric structure, where feature embeddings of augmented samples belonging to the same image tend to cluster around their respective means. Interestingly, at later stages of training, a similar trend appears with respect to semantic classes.

- **The role of regularization:** In Figure 3 (left) we show that the accuracy of extracting (i.e., linear probing) semantic classes from SSL-trained representations continuously improves even after the model accurately clusters the augmented training samples based on their sample identity. As can be seen, the regularization constraint plays a key role in inducing the clustering of data into semantic attributes at the embedding space.

- **Impact of randomness:** We argue that SSL-trained representation functions are highly correlated with semantic classes. To support this claim, we study how the degree of randomness in the targets affects the ability to learn them from a pretrained representation. As we show in Figure 4, the alignment between the targets and the learned representations substantially improves as the targets become less random and more semantically meaningful.

- **Learning hierarchic representations:** We study how SSL algorithms learn representations that exhibit hierarchic classes. We demonstrate that throughout the training process, the ability to cluster and distinguish between samples with respect to different levels of semantic targets continually improves at all layers of the model. We consider several levels of hierarchy, such as the sample level (the sample identity), semantic classes (such as different types of animals and vehicles), and high-level classes (e.g., animals, and vehicles).

  Furthermore, we analyze the ability of each layer to capture distinct semantic classes. As shown in Figure 5, we observe that the clustering and separation ability of each layer improves as we move deeper into the network. This phenomenon closely resembles the behavior observed in supervised learning [10, 26, 56, 62], despite the fact that SSL-trained models are trained without direct access to the semantic targets.

In Section 4, we investigate various clustering properties in the top layer of SSL-trained networks. In Section 5, we conduct several experiments to understand what kinds of functions are encoded in SSL-trained neural networks and can be extracted from their features. In Section 6, we study how SSL algorithms learn different hierarchies and how this occurs at intermediate layers.

## 2 Background and Related Work

### 2.1 Self-Supervised Learning

SSL is a family of techniques that leverages unlabeled data to learn representation functions that can be easily adapted to new downstream tasks. Unlike supervised learning, which relies on labeled data, SSL employs self-defined signals to establish a proxy objective. The model is pre-trained using this proxy objective and then fine-tuned on the supervised downstream task. However, a major challenge of this approach is to prevent 'representation collapse', where the model maps all inputs to the same output [39].

Several methods have been proposed to address this challenge. *Contrastive learning methods* such as SimCLR [14] and its InfoNCE criterion [64] learn representations by maximizing the agreement between positive pairs (augmentations of the same sample) and minimizing the agreement between negative pairs (augmentations of different samples). In contrast, *non-contrastive learning methods* [6, 66, 34] replace the dependence on contrasting positive and negative samples, by applying certain regularization techniques to prevent representation collapse. For instance, some methods use stop-gradients and extra predictors to avoid collapse [16, 34], while others use an additional clustering constraints [13].

**Variance-Invariance-Covariance Regularization (VICReg).** A widely used method for SSL training [6], which generates representations for both an input and its augmented counterpart. It aims to optimize two key aspects:

- *Invariance loss* - The mean-squared Euclidean distance between pairs of embedding, serving to ensure consistency between the representation of the original and augmented inputs.

- *Regularization* - Comprising two elements, the **variance loss**, which promotes increased variance across the batch dimension through a hinge loss restraining the standard deviation, and the **covariance loss**, which penalizes off-diagonal coefficients of the covariance matrix of the embeddings to foster decorrelation among features.

This leads to the VICReg loss function:

$$L(f) = \underbrace{\lambda s(Z, Z')}_{\text{Invariance}} + \underbrace{\mu[v(Z) + v(Z')] + \nu[c(Z) + c(Z')]}_{\text{Regularization}}, \tag{1}$$

where $L(f)$ is minimized over batches of samples $I$. $s(Z, Z')$, $v(Z)$, and $c(Z)$ denote the invariance, variance, and covariance losses respectively, and $\lambda, \mu, \nu > 0$ are hyperparameters controlling the balance between these loss components.

**A Simple Framework for Contrastive Learning of Visual Representations (SimCLR).** The SimCLR method [14], another popular approach, minimizes the contrastive loss function [35]. Given two batches of embeddings $Z, Z'$, we can also decompose the objective into an invariance term and a

regularization term:

$$L(f) = -\frac{1}{B} \sum_{i=1}^{B} \underbrace{(\mathrm{sim}(Z_i, Z_i')/\tau)}_{\text{Invariance}} - \underbrace{\log \sum_{j \neq i} \exp(\mathrm{sim}(Z_i, Z_j')/\tau)}_{\text{Regularization}},$$

where $\mathrm{sim}(a, b) := \frac{a^\top b}{\|a\| \cdot \|b\|}$ is the cosine similarity between vectors $a$ and $b$, and $\tau > 0$ is a 'temperature' parameter that controls the sharpness of the distribution. Intuitively, minimizing the loss encourages the representations $Z_i$ and $Z_i'$ of the same input sample to be similar (i.e., have a high cosine similarity) while pushing away the representations $Z_i$ and $Z_j'$ of different samples (i.e., have a low cosine similarity) for all $i, j \in [B]$.

## 2.2  Neural Collapse and Clustering

In a recent paper [50], it was demonstrated that deep networks trained for classification tasks exhibit a notable behavior: the top-layer feature embeddings of training samples of the same class tend to concentrate around their respective class means, which are maximally distant from each other. This behavior is generally regarded as desirable since max-margin classifiers tend to exhibit better generalization guarantees (e.g., [2, 7, 33]), and clustered embedding spaces are useful for few-shot transfer learning (e.g., [32, 28, 27, 29]).

In this paper, we aim to explore whether similar clustering tendencies occur in SSL-trained representation functions at both the sample level (with respect to augmentations) and at the semantic level. To address this question, we first introduce some key aspects of neural collapse. Specifically, we will focus on class-feature variance collapse and nearest class-center (NCC) separability.

To measure these properties, we use the following metrics. Suppose we have a representation function $f : \mathbb{R}^d \to \mathbb{R}^p$ and a collection of unlabeled datasets $S_1, \ldots, S_C \subset \mathbb{R}^d$ (each corresponding to a different class). The feature-variability collapse measures to what extent the samples from the same class are mapped to the same vector. To quantify this property we use the averaged **class-distance normalized variance (CDNV)** [28] $\mathrm{Avg}_{i \neq j}[V_f(S_i, S_j)] = \mathrm{Avg}_{i \neq j}\left[\frac{\mathrm{Var}_f(S_i) + \mathrm{Var}_f(S_j)}{2\|\mu_f(S_i) - \mu_f(S_j)\|^2}\right]$, where $\mu_f(S)$ and $\mathrm{Var}_f(S)$ are the mean and variance of the uniform distribution $U[\{f(x) \mid x \in S\}]$.

A weaker notion of clustering is the nearest class-center (NCC) separability [26], which measures to what extent the feature embeddings of training samples form centroid-like geometry. The NCC separability asserts that, during training, the penultimate-layer's feature embeddings of training samples can be accurately classified with the 'nearest class-center (NCC) classifier', $h(x) = \arg\min_{c \in [C]} \|f(x) - \mu_f(S_c)\|$. To measure this property, we compute the NCC train and test accuracies, which are simply the accuracy rates of the NCC classifier.

A recent paper [22] argued that idealized SSL-trained representations simultaneously cluster data into multiple equivalence classes. This implies that there are various ways to categorize the data that result in high NCC accuracy. However, achieving a very small CDNV for multiple different categorizations is impossible since a zero CDNV would mean that feature embeddings of two samples from the same category are identical. Consequently, an intriguing question arises: with respect to which sets of classes do the learned feature embeddings cluster?

## 2.3  Reverse Engineering Neural Networks

Reverse engineering neural networks have recently garnered attention as an approach to explain how neural networks make predictions and decisions. While substantial work has been dedicated to understanding the functionalities of intermediate layers of neural network classifiers [23, 9, 17, 50, 51, 60, 56, 3], characterizing the functionalities of representation functions generated by SSL algorithms remains an open problem.

A major source of complexity is the reliance on a pretext task and image augmentations. For instance, to fully understand the success of SSL it is essential to understand the relationship between the pretext task and the downstream task [58, 65]. It is yet unclear how to properly train representations with SSL, and therefore, SSL algorithms are quite complicated and include many different types of losses [14, 66, 6] and optimization tricks [14, 16, 37]. Of note, it is not obvious what are the differences between representations that are learned with contrastive learning and non-contrastive

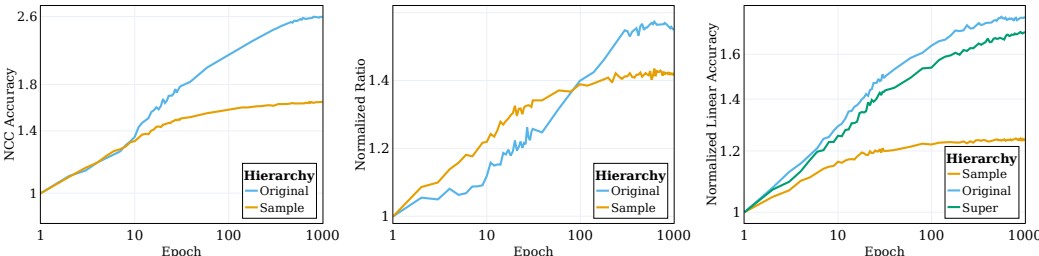

Figure 2: **SSL algorithms cluster the data with respect to semantic targets. (left)** The normalized NCC train accuracy, computed by dividing the accuracy values by their value at initialization. **(middle)** The ratio between the NCC test accuracy and the linear test accuracy for per-sample and semantic classes normalized by its value at initialization. **(right)** The linear test accuracy rates, normalized by their values at initialization. All experiments are conducted on CIFAR-100 with VICReg training.

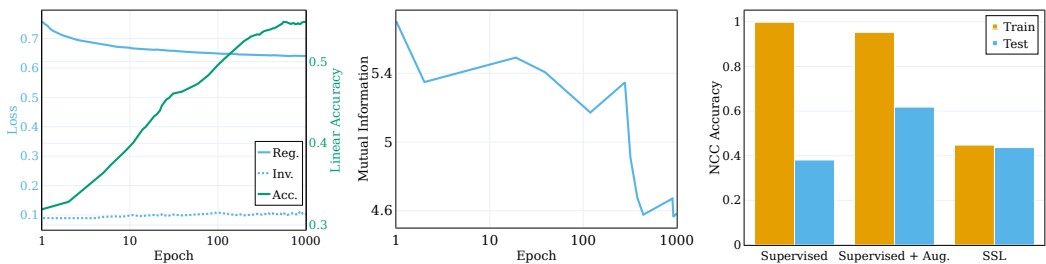

Figure 3: **(left)** The regularization and invariance losses together with the original target linear test accuracy of an SSL-trained model over the course of training. **(middle)** Compression of the mutual information between the input and the representation during the course of training. **(right)** SSL training learns clustered representations. The NCC train and test accuracy in supervised (with/without augmentation), and SSL. All experiments are conducted on CIFAR-100 with VICReg training.

learning methods [52]. In this work, we make new strides to understand SSL-trained representations, by investigating their clustering properties with respect to various types of classes.

## 3 Problem Setup

Self-supervised learning (SSL) is frequently used to pretrain models, preparing them for adaptation to new downstream tasks. This raises an essential question: How does SSL training influence the learned representation? Specifically, what underlying mechanisms are at work during this training, and which classes can be learned from these representation functions? To investigate this, we train SSL networks across various settings and analyze their behavior using different techniques. All the training details can be found in Appendix A.

**Data and augmentations.** Throughout all of the experiments (in the main text) we used the CIFAR-100 [41] image classification dataset, which contains 100 classes of semantic objects (original classes), which are also divided into 20 superclasses, with each superclass containing 5 classes (for example, the 'fish' superclass contains the 'aquarium fish', 'flatfish', 'ray', 'shark' and 'trout' categories). In order to train our models we used the image augmentation protocol introduced in SimCLR [14]. Each SSL training session is carried out for 1000 epochs, using the SGD optimizer with momentum. Usually, SSL-trained models are tested on a variety of downstream datasets to assess their performance. To simplify our analysis and minimize the influence of potential distribution shifts, we specifically evaluate the models' performance on the CIFAR-100 dataset. Such evaluations are typically done in SSL research [6, 34, 66], usually training on and evaluating on ILSVRC (ImageNet). Additional evaluations on various datasets (CIFAR-10 [41], FOOD101 [11], Aircrafts [45], Tiny Imagnet [43], and Imagenet [19]) can be found in Appendix B.2.

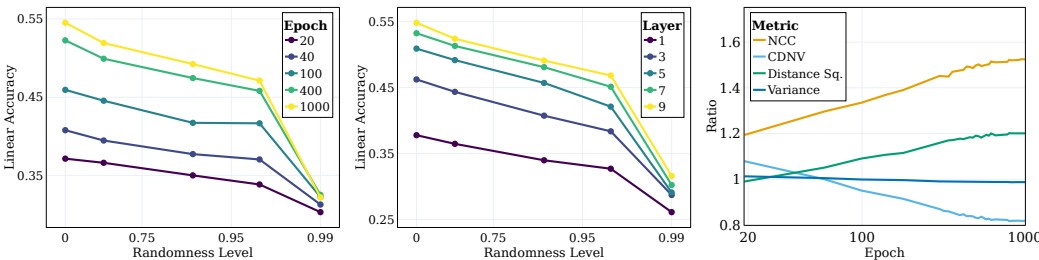

Figure 4: **SSL continuously learns semantic targets over random ones. (left)** The linear test accuracy for targets with varying levels of randomness from the last layers at different epochs. **(middle)** The linear test accuracy for targets with varying levels of randomness for the trained model. **(right)** The ratios between non-random and random targets for various clustering metrics. All experiments are conducted on CIFAR-100 with VICReg training.

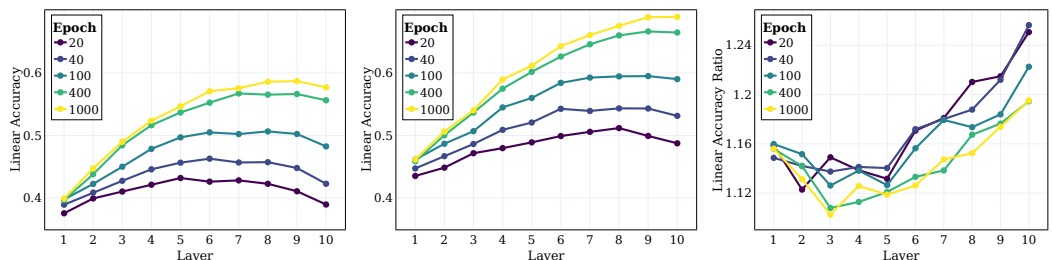

Figure 5: **SSL efficiently learns semantic classes throughout intermediate layers.** The linear test accuracy of different layers of the model at various epochs **(left)** With respect to the 100 original classes. **(middle)** With respect to the 20 superclasses. **(right)** The ratio between the superclass and the original classes. All experiments are conducted on CIFAR-100 with VICReg training.

**Backbone architecture.** In all our experiments, we utilized the RES-$L$-$H$ architecture as the backbone, coupled with a two-layer multi-layer perceptron (MLP) projection head. The RES-$L$-$H$ architecture is a variant of the ResNet architecture [38] where the width $H$ remains constant across all $L$ residual blocks as introduced in [26].

**Linear probing.** To evaluate the effectiveness of extracting a given discrete function (e.g., categories) from a representation function, we employ linear probing. In this approach, we train a linear classifier, also known as a linear probe, on top of the representation, using a set of training samples. To train the linear classifier, we used the Linear Support Vector Classification (LSVC) algorithm from the scikit-learn library [54]. The "linear accuracy" is determined by measuring the performance of the trained linear classifier on a validation set.

**Sample level classification.** We aim to evaluate the capacity of a given representation function to correctly classify different augmentations originating from the same image and distinguish them from augmentations of different images. To facilitate this, we constructed a new dataset specifically designed to evaluate sample-level separability.

The training dataset comprises 500 random images sourced from the CIFAR-100 training set. Each image represents a particular class and undergoes 100 different augmentations. As such, the training dataset contains 500 classes with a total of 50000 samples. For the test set, we utilize the same 500 images but apply 20 different augmentations drawn from the same distribution. This results in a test set consisting of 10000 samples. To measure the linear or NCC accuracy rates of a given representation function at the sample level, we first compute a relevant classifier using the training data and subsequently evaluate its accuracy rates on the corresponding test set.

## 4 Unraveling the Clustering Process in Self-Supervised Learning

The clustering process has long played a significant role in aiding the analysis of deep learning models [1, 17, 50]. To gain intuition behind the SSL training, in Figure 1 we present a UMAP [57]

visualization of the network's embedding space for the training samples, both pre-and post-training, across different hierarchies.

As anticipated, the training process successfully clusters together samples at the per-sample level, mapping different augmentations of the same image (as depicted in the first row). This outcome is unsurprising given that the objective inherently encourages this behavior (through the invariance loss term). Yet, more remarkably, the training process also clusters the original 'semantic classes' from the standard CIFAR-100 dataset, even though the labels were absent during the training process. Interestingly, the high-level hierarchy (the superclasses) is also effectively clustered. This example demonstrates that while our training procedure directly encourages clustering at the sample level, the SSL-trained representations of data are additionally clustered with respect to semantic classes across different hierarchies.

To quantify this clustering process further, we train a RES-10-250 network using VICReg. We measure the NCC train accuracy, which is a clustering measure (as described in Section 2.2), both at the sample level and with respect to the original classes. Notably, the SSL-trained representation exhibits neural collapse at a sample level (with an NCC train accuracy near $1.0$), yet the clustering with respect to the semantic classes is also significant ($\approx 0.41$ on the original targets).

As shown in Figure 2 (left), most of the clustering process with respect to the augmentations (on which the network was directly trained) occurs at the beginning of the training process and then plateaus, while the clustering with respect to semantic classes (which is not directly featured in the training objective) continues to improve throughout the training process.

A key property observed in [50] is that the top-layer embeddings of supervised training samples gradually converge towards a centroid-like structure. In our quest to better understand the clustering properties of SSL-trained representation functions, we investigated whether something similar occurs with SSL. The NCC classifier's performance, being a linear classifier, is bounded by the performance of the best-performing linear classifier. By evaluating the ratio between the accuracy of the NCC classifier and a linear classifier trained on the same data, we can investigate data clustering at various levels of granularity. In Figure 2 (middle), we monitor this ratio for the sample-level classes and for the original targets, normalized with respect to its value at initialization. As SSL training progresses, the gap between NCC accuracy and linear accuracy narrows, indicating that augmented samples progressively cluster more based on their sample identity and semantic attributes. Furthermore, the plot shows that the sample-level ratio is initially higher, suggesting that augmented samples cluster based on their identity until they converge to centroids (the ratio between the NCC accuracy and the linear accuracy is $\geq 0.9$ at epoch 100). However, as training continues, the sample-level ratio saturates while the class-level ratio continues to grow and converges to $\approx 0.75$. This implies that the augmented samples are initially clustered with respect to their sample identities, and once achieved, they tend to cluster with respect to high-level semantic classes.

**Implicit information compression in SSL training.** Having examined the clustering process within SSL training, we next turn our attention to analyzing information compression during the learning process. As outlined in Section 2.2, effective compression often yields representations that are both beneficial and practical. Nevertheless, it is still largely uncharted territory whether such compression indeed occurs during the course of SSL training [59].

To shed light on this, we utilize the Mutual Information Neural Estimation (MINE) [8], a method designed to estimate the mutual information between the input and its corresponding embedded representation during the course of training. This metric serves as an effective gauge of the representation's complexity level, essentially demonstrating how much information (number of bits) it encodes.

In Figure 3 (middle) we report the average mutual information computed across 5 different MINE initialization seeds. As demonstrated, the training process manifests significant compression, culminating in highly compact trained representations. This observation aligns seamlessly with our previous findings related to clustering, where we noted that more densely clustered representations correspond to more compressed information.

**The role of regularization loss.** To gain a deeper understanding of the factors influencing this clustering process, we study the distinct components of the SSL objective function. As we discussed in section 2.1, the SSL objective function is composed of two terms: invariance and regularization. The invariance term's main function is to enforce similarity among the representations of augmentations of the same sample. In contrast, the regularization term helps to prevent representation collapse.

To investigate the impact of these components on the clustering process, we dissect the objective function into the invariance and regularization terms and observe their behavior during the course of training. As part of this exploration, we trained a RES-5-250 network using VICReg with $\mu = \lambda = 25$ and $\nu = 1$, as proposed in [6]. Figure 3 (left) presents this comparison, charting the progression of loss terms and the linear test accuracy with respect to the original semantic targets. Contrary to popular belief, the invariance loss does not significantly improve during the training process. Instead, the improvement in loss (and downstream semantic accuracy) is achieved due to minimizing the regularization loss. This observation is consistent with our earlier finding that the per-sampling clustering process (which is driven by the invariance loss) saturates early in the training process.

We conclude that the majority of the SSL training process is geared towards improving the semantic accuracy and clustering of the learned representations, rather than the per-sample classification accuracy and clustering. This is consistent with the observed trend of the regularization loss improving over the course of training and the early saturation of the invariance loss.

In essence, our findings suggest that while self-supervised learning directly targets sample-level clustering, the majority of the training time is spent on orchestrating data clustering according to semantic classes across various hierarchies. This observation underscores the remarkable versatility of SSL methodologies in generating semantically meaningful representations through clustering and sheds light on its underlying mechanisms.

**Comparing supervised learning and SSL clustering.**  As mentioned in Section 4, deep network classifiers tend to cluster training samples into centroids based on their classes. However, the learned function is truly clustering the classes only if this property holds for the test samples, which is expected to occur but to a lesser degree.

An interesting question is to what extent SSL implicitly clusters the samples according to their semantic classes, compared to the clustering induced by supervised learning. In Figure 3 (right), we report the NCC train and test accuracy rates at the end of training for different scenarios: supervised learning with and without augmentations and SSL. For all cases, we used the RES-10-250 architecture.

While the NCC train accuracy of the supervised classifier is 1.0, which is significantly higher than the NCC train accuracy of the SSL-trained model, the NCC test accuracy of the SSL model is slightly higher than the NCC test accuracy of the supervised model. This implies that both models exhibit a similar degree of clustering behavior with respect to the semantic classes. Interestingly, when training the supervised model with augmentation slightly decreases the NCC train accuracy, yet significantly improves the NCC test accuracy.

## 5  Exploring Semantic Class Learning and the Impact of Randomness

Semantic classes define relationships between inputs and targets based on inherent patterns in the input. On the other hand, mappings from inputs to random targets lack discernible patterns, resulting in seemingly arbitrary connections between inputs and targets.

In this section, we study the influence of randomness on a model's proficiency in learning desired targets. To do so, we build a series of target systems characterized by varying degrees of randomness and examine their effect on the learned representations. We train a neural network classifier on the same dataset for classification and use its target predictions at different epochs as targets with varying degrees of randomness. At epoch 0, the network is random, generating deterministic yet seemingly arbitrary labels. As training advances, the functions become less random, culminating in targets that align with ground truth targets (considered non-random). We normalize the degree of randomness between 0 (non-random, end of training) and 1 (random, initialization). Utilizing this methodology, we investigate the classes SSL learns during training.

Initially, our emphasis is on exploring the impact of the SSL training process on the learned targets. We train a RES-5-250 network with VICReg and employ a ResNet-18 to generate targets with different levels of randomness. Figure 4 (left) showcases the linear test accuracy for varying degrees of random targets. Each line corresponds to the accuracy at a different stage of SSL training for different levels of randomness. As we can see, the model captures classes closer to "semantic" ones (lower degrees of randomness) more effectively throughout training while not showing significant performance improvement on highly random targets. For results with additional random target generators, see Appendix B.5.

A key question in deep learning revolves around understanding the functionalities and the impact of intermediate layers on classifying different types of classes. For example, do different layers learn different kinds of classes? We explore this by evaluating the linear test accuracy of different layer representations for various degrees of target randomness at the end of the training. As shown in Figure 4 (middle), the linear test accuracy consistently improves at all layers as randomness decreases, with deeper layers outperforming across all class types and the performance gap broadening for classes closer to semantic ones.

Beyond classification accuracy, a desirable representation exhibits a high degree of clustering with respect to the targets [50, 24, 32, 28]. We assess the quality of clustering using several metrics: NCC accuracy, CDNV, average per-class variance, and average squared distance between class means [28]. To gauge the improvement in representation over training, we calculate the ratios of these metrics for semantic vs. random targets. Figure 4(right) displays these ratios, indicating that the representation increasingly clusters the data with respect to semantic targets compared to random ones, as the ratio of the NCC, CDNV, increases during training. Interestingly, we see that the decrease in the CDNV, which is the variance divided by the squared distance, is caused solely by the increase in squared distance. The variance ratio stays fairly constant during training. This phenomenon of encouraging large margins between clusters has been shown to improve the performance [24, 17].

## 6 Learning Class Hierarchies and Intermediate Layers

Previous studies have given evidence that in the context of supervised learning, intermediate layers gradually capture features at varying levels of abstraction [1, 17, 50, 26, 10]. The initial layers tend to focus on low-level features, while deeper layers capture more abstract features. In Section 4, we showed that SSL implicitly learns representations highly correlated with semantic attributes. Next, we investigate whether the network learns higher-level hierarchical attributes and which layers are better correlated with these attributes.

In order to measure what kinds of targets are associated with the learned representations, we trained a RES-5-250 network using VICReg. We compute the linear test accuracy (normalized by its value at initialization) with respect to three hierarchies: the sample-level, the original 100 classes, and the 20 superclasses. In Figure 2 (right), we plot these quantities computed for the three different sets of classes. We observed that, in contrast to the sample-level classes, the performance with respect to both the original classes and the superclasses significantly increased during training. The complete details are provided in Appendix A.

As a next step, we investigate the behavior of intermediate layers of SSL-trained models and their ability to capture targets of different hierarchies. To this end, we trained a RES-10-250 with VICReg. In Figure 5 (left and middle), we plot the linear test accuracy across all intermediate layers at various stages of training, measured for both the original targets and the superclasses. In Figure 5 (right) we plot the ratios between the superclasses and the original classes.

We draw several conclusions from these results. First, we observe a consistent improvement in clustering as we move from earlier to deeper layers, becoming more prominent during the course of training. Furthermore, similar to supervised learning scenarios [62, 26, 10], we find that the linear accuracy of the network improves in each layer during SSL training. Of note, we find that for the original classes, the final layers are not the optimal ones. Recent studies in SSL [49, 55] showed that the performance of the different algorithms are highly sensitive to the downstream task domains. Our study extends this observation and suggests that also different parts of the network may be more suitable for certain downstream tasks and class hierarchies. Figure 5(right), it is evident that relatively, the accuracy of the superclasses improves more than that of the original classes in the deeper layers of the network.

## 7 Conclusions

Representation functions that cluster data based on semantic classes are generally favored due to their ability to classify classes accurately with limited samples [32, 28, 27, 29]. In this paper, we conduct a comprehensive empirical exploration of SSL-trained representations and their clustering properties concerning semantic classes. Our findings reveal an intriguing impact of the regularization constraint in SSL algorithms. While regularization is primarily used to prevent representation collapse, it also enhances the alignment between learned representations and semantic classes, even after accurately clustering augmented training samples. We investigate the emergence of different class types during

training, including targets with various hierarchies and levels of randomness, and examine their alignment at different intermediate layers. Collectively, these results provide substantial evidence that SSL algorithms learn representations that effectively cluster based on semantic classes.

Despite the similarities observed between the supervised setting and the SSL setting in our paper, several questions remain unanswered. Foremost among them is the inquiry into why SSL algorithms learn semantic classes. Although we present compelling evidence linking this phenomenon to the regularization constraint, the explanation of how this term manages to cluster data with respect to semantic classes, and the specific types of classes being learned, remains unclear. A deeper understanding of the types of classes to which data clusters, coupled with the connection between neural collapse and transfer learning [28, 27, 29], may be helpful in understanding the transferability of SSL-trained representations to downstream tasks.

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

## A    Training Details

### A.1    Data augmentation

We follow the image augmentation protocol first introduced in SimCLR [14] and now commonly used by similar approaches based on siamese networks [13, 34, 14, 66]. Two random crops from the input image are cropped and resized to $32 \times 32$, followed by random horizontal flip, color jittering of brightness, contrast, saturation and hue, Gaussian blur, and random grayscale. Each crop is normalized in each color channel using the ImageNet mean and standard deviation pixel values. The following operations are performed sequentially to produce each view:

- Random cropping with an area uniformly sampled with size ratio between 0.08 to 1.0, followed by resizing to size $32 \times 32$. `RandomResizedCrop(32, scale=(0.08, 1.0))` in PyTorch.

- Random horizontal flip with probability 0.5.

- Color jittering of brightness, contrast, saturation and hue, with probability 0.8. `ColorJitter(0.4, 0.4, 0.2, 0.1)` in PyTorch.

- Grayscale with probability 0.2.

- Gaussian blur with probability 0.5 and kernel size 23.

- Solarization with probability 0.1.

- Color normalization with mean (0.485, 0.456, 0.406) and standard deviation (0.229, 0.224, 0.225) (ImageNet standardization).

**Supervised learning - Figure 3 (right).**    For the supervised learning setting, we used AutoAugment [18] with a policy created for CIFAR-10 [41].

### A.2    Network Architectures

In this section, we describe the network architectures used in our experiments.

**SSL backbone.**    The main architecture used as the SSL backbone is a convolutional residual network, denoted RES-$L$-$H$. It consists of a stack of two $2 \times 2$ convolutional layers with stride 2, batch normalization, and ReLU activation, followed by $L$ residual blocks. The $i$th block computes $g^i(x) = \sigma(x + B_i^2(C_i^2(\sigma(B_i^1(C_i^1(x))))))$, where $C_i^j$ is a $3 \times 3$ convolutional layer with $H$ channels, stride 1, and padding 1, $B_i^j$ is a batch normalization layer, for $j \in \{1, 2\}$, and $\sigma$ is the ReLU activation.

**Random target functions.**    Throughout the paper, we use two different random target functions, a ResNet-18 [38] and a visual transformer (ViT) [21] (Appendix B.5). For the transformer architecture, we used 10 layers, with 8 attention heads and 384 hidden dimensions. The classification was done using a classification token.

### A.3    Optimization

**SSL.**    We trained all of our SSL models for 1000 epochs using a batch size of 256. We used the SGD optimizer with a learning rate of 0.002, a momentum value of 0.9, and a weight decay value of $1e{-}6$. Additionally, we used a CosineAnnealing learning rate scheduler [44].

**Random targets.**    The ResNet-18 targets were trained using the Cross-Entropy loss, with the Adam optimizer [40], using a learning rate of 0.05 and the Cosine Annealing learning rate scheduler. For the ViT training, we used the Adam optimizer with a 0.001 learning rate, weight decay of 5e-5, and a Cosine Annealing learning rate scheduler. We used the model's class predictions as the targets.

### A.4    Implementation Details

Our experiments were implemented in PyTorch [53], utilizing the Lightly [61] library for SSL models and PyTorch Lightning [25] for training. All of our models were trained on V100 GPUs.

## B    Additional Experiments

**Clustering in SSL.**    In Figure 2 in the main text, we reported behaviors of extracting classes of different hierarchies from SSL-trained models. In this experiment, we provide the un-normalized version of the results in Figure 2. Specifically, in Figure 6 (left), we show the un-normalized results, along with the normalized plots shown in the main text Figure 6 (right).

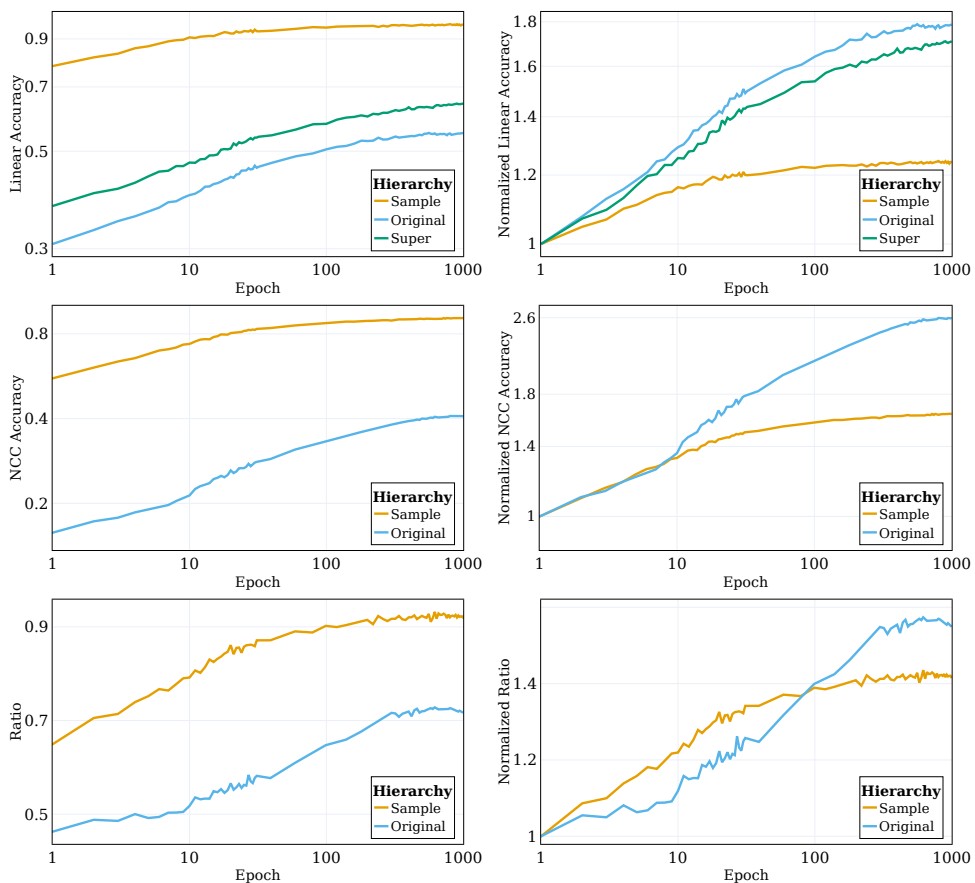

Figure 6: **SSL algorithms cluster the data with respect to semantic targets. (right)** The results in Figure 2. **(left)** The unnormalized version of the same results.

## B.1 Network Architectures

SSL research primarily focuses on utilizing a single ResNet-50 backbone for experimental purposes. However, given the similarities between SSL and supervised learning, as demonstrated in the paper, it is worth exploring how the choice of backbone architecture affects the clustering of representations with respect to semantic targets. In Figure 7, we present the intermediate layer clustering for different network architectures for extracting the original classes (left) and the superclasses (right). In Figure 7 (top), we display the linear test accuracy of RES-5-50, RES-5-250, and RES-5-1000 networks at different epochs (20, 40, 100, 400, 1000). As can be seen, the network's width has a significant impact on the results for all intermediate layers and at all training epochs.

In Figure 7 (bottom), we present the linear test accuracy of networks RES-5-250, RES-10-250 throughout the training epochs (left) and intermediate layers (right). It is generally seen that in the later epochs of training, the additional layers benefit the linear test performance of the network. However, it is also interesting to note that for the initial layers of the network, the shallow RES-5-250 model gets more clustered representations both for the original classes and for the superclasses. This hints at the fact that deeper networks may need intermediate layer losses to encourage clustering in initial layers [24, 10].

**Experiments with Vision Transformers.** We extend our experimentation to different backbones to validate the results illustrated in the main paper. A Vision Transformer (ViT) is adopted as per the architecture detailed in [20], employing a 5-layer transformer with 4 heads and a hidden dimension of 384. In Figure 8, we examine the clustering degree produced by various models trained using different SSL algorithms on the Tiny Imagenet dataset. Specifically, we report the NCC train accuracy across intermediate layers for both the VICReg and SimCLR algorithms.

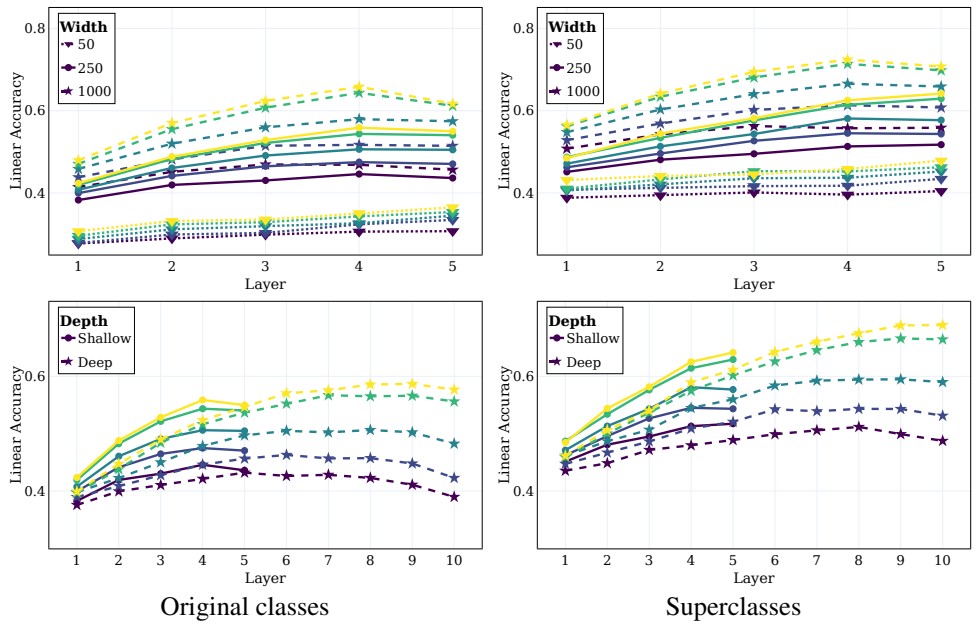

Figure 7: **The influence of width and depth on learning semantic classes at intermediate layers.** **(top)** Linear test accuracy at different epochs for neural networks of varying widths. **(bottom)** Linear test accuracy of neural networks with different depths. **(left)** The performance is measured in relation to the original classes. **(right)** The performance with respect to the superclasses.

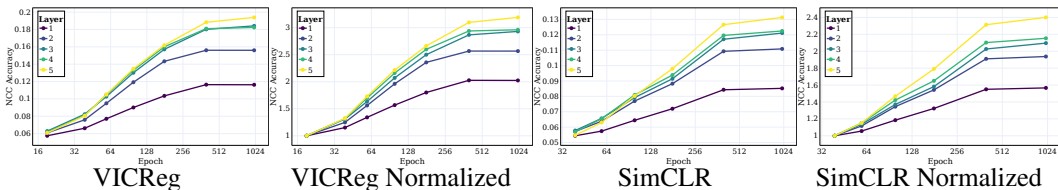

Figure 8: **VICReg and SimCLR with ViT [20] backbone clusters the data with respect to semantic targets in intermediate layers.** The NCC train accuracy (normalized and un-normalized) of an SSL-trained network on **Tiny Imagenet**, measured at intermediate layers.

### B.2 Experiments with Additional Datasets

**CIFAR-10.** We run similar experiments using RES-5-250 on the CIFAR-10 dataset. In Figure 10 (top), we report the NCC test accuracy for recovering both the sample level and the class labels. Similar to the results in the main text, the samples progressively cluster around their augmentation class means early in training. However, during the majoring of the training process, the augmented samples tend to cluster with respect to the semantic labels. In Figure 10 (bottom), we report the NCC and linear (from left to right) test accuracies of intermediate layers with respect to the semantic labels.

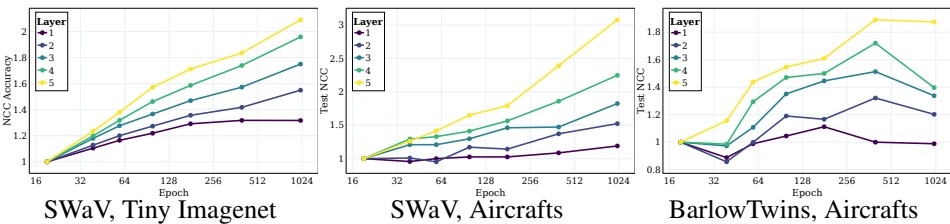

Figure 9: **Different algorithms cluster the data with respect to semantic targets in intermediate layers.** The (normalized) NCC train accuracy of an SSL-trained RES-5-250 network on different datasets, measured at intermediate layers.

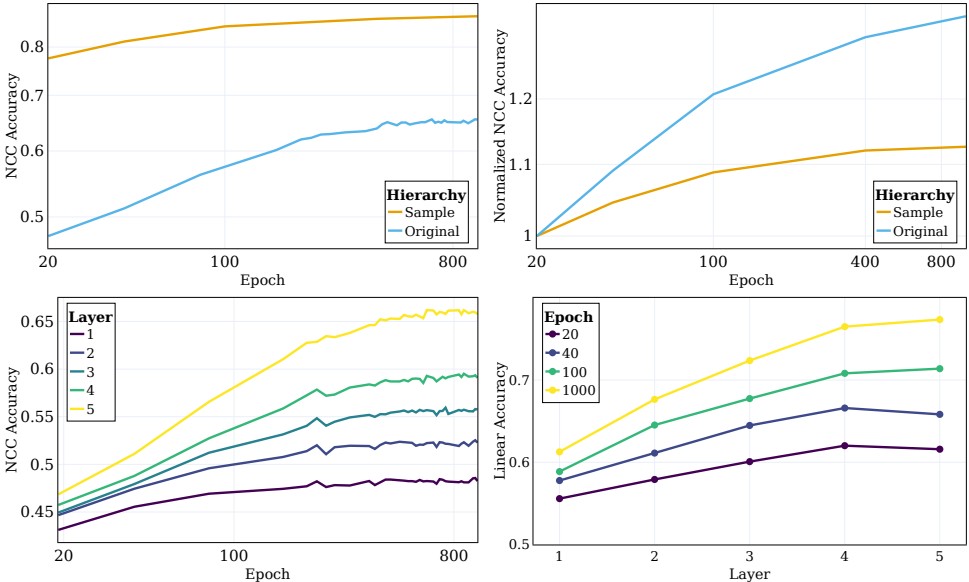

Figure 10: **SSL algorithms cluster the data with respect to semantic targets. (top)** The NCC train accuracy of an SSL-trained network on CIFAR-10, measured at the sample level and original classes (both un-normalized and normalized). **(bottom)** The NCC train accuracy of the model at different layers and epochs.

As can be seen, the degree of clustering monotonically improves at all layers during the course of training.

**FOOD-101.** Similar to CIFAR-10, we experiment on the FOOD101 dataset, consisting of 75,750 train samples and 25,250 test samples across 101 food classes (e.g. 'pizza', 'misso soup', 'waffles'), using VICReg and the RES-5-250 architecture. In Figure 13 (left), we plot the NCC test accuracy for both the sample level, and the semantic classes level, and in Figure 13 (right) we plot the NCC test accuracy of the intermediate layers with respect to the sample level labels. The results are similar to the ones shown in Section 4. Additionally, we present the sample level clustering in the intermediate layers. Here, we also see an improvement throughout the layers and the epochs.

**Aircrafts.** We applied the SWaV and Barlow Twins algorithms with the RES-5-250 architecture to the Aircrafts dataset, which consists of 10,000 images from 100 aircraft types. In Figure 9 we show the NCC train accuracy at the various intermediate layers.

**Tiny Imagenet.** We also experiment on more varied datasets, like the Tiny ImageNet dataset, consisting of 100,000 train samples and 10,000 test samples across 200 classes (e.g. 'nematode', 'harmonica', 'syringe'), using VICReg and the RES-10-250 architecture. In Figure 12 we plot the NCC test accuracy for both the sample level, and the semantic classes level, both unormalized and normalized resp. In Figure 9 we show the NCC train accuracy using the SWav SSL algorithm. The results are similar to the ones shown in the clustering section.

**Imagenet.** We extend our experimentation to the more complex ImageNet [19] dataset. In Figure 11, we show the linear test accuracy and NCC train accuracy of both SwAV and SimCLR on the ImageNet dataset, along training epochs, with respect to the semantic labels on the final layer of a ResNet-50. Similarly to the supervised setting, as shown in section Section 4, both the linear accuracy and the NCC accuracy dramatically improves during training.

### B.3 Hyperparameters

In Figure 14, we present the training losses and linear accuracies for three different hyperparameter selections trained using VICReg and a RES-5-250 network. Specifically, we vary the $\mu$ (variance regularization) parameter and display the losses and linear accuracy for $\mu = 5, 25, 100$, from left to

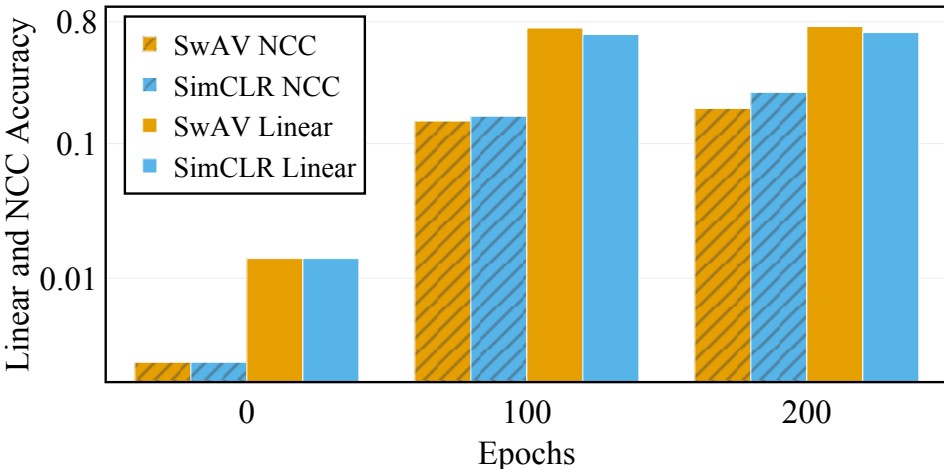

Figure 11: **SwAV and SimCLR cluster ImageNet data with respect to semantic targets.** The last layer Linear Test accuracy and NCC train accuracy of an SSL-trained ResNet-50 on **Imagenet**, on the 1000 original classes.

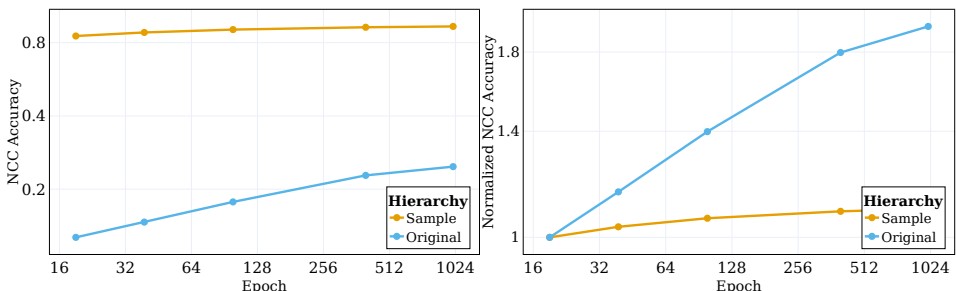

Figure 12: **VICReg clusters the data with respect to semantic targets** The NCC train accuracy of an SSL-trained RES-10-250 on **Tiny Imagenet**, measured at the sample level and original classes (both un-normalized and normalized).

right, respectively. Both loss terms are normalized by $\mu$. In all cases, we used $\nu = 1$ and $\lambda = 25$ by default.

Interestingly, the configuration with $\mu = 5$ exhibits significantly lower performance compared to the default setting ($\mu = 25$). However, the configuration with $\mu = 100$ achieves performance comparable to the default, despite the network having a considerably higher invariance loss term. This observation further supports our claim that the regularization term plays a crucial role in learning semantic features.

### B.4 Experiments with Different SSL Algorithms

In our main text, we primarily focus on the VICReg SSL algorithm. However, to ensure the robustness of our findings across different algorithms, we also trained a network using the SimCLR algorithm [14] with our default RES-5-250 architecture. In Figure 15, we present several experiments that compare the network trained with VICReg to the one trained with SimCLR.

Firstly, Figure 15 (top) illustrates the linear test accuracies of intermediate layers for both algorithms throughout the training epochs with respect to the original classes (left) and the superclasses (right). The darkness of the lines indicates the progression of epochs, with lighter shades representing later epochs. Despite the significant differences between the algorithms, particularly in SimCLR's contrastive nature, we observe similar clustering behavior across different layers and epochs with VICReg consistently achieving better performance. This finding demonstrates that various SSL algorithms yield comparable clustering characteristics.

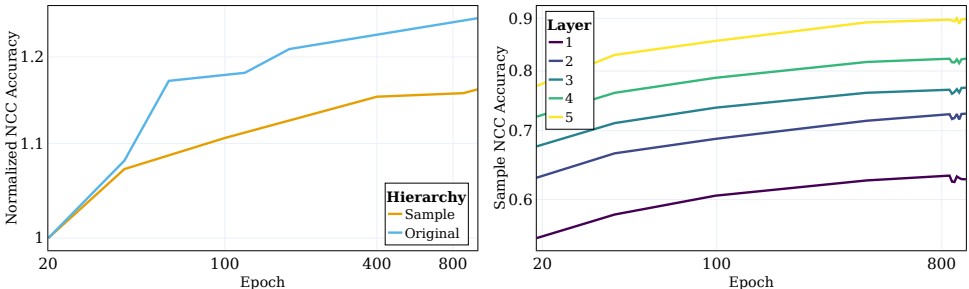

Figure 13: **SSL algorithms cluster the data with respect to semantic targets and invariance to augmentations at intermediate layers. (left)** The normalized NCC test accuracy of a VICReg-trained network on FOOD101 with respect to the sample level labels and the original class labels. **(right)** The NCC test accuracy of the model with respect to the sample level labels of intermediate layers.

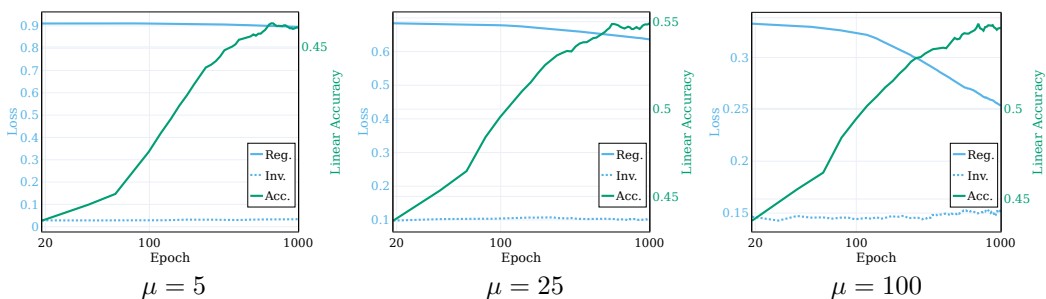

Figure 14: **The role of the regularization term in SSL training.** Each plot depicts the regularization and invariance losses, along with the linear test accuracy, throughout the training process of VICReg with $\mu = 5, 25, 100$ respectively.

Furthermore, in addition to clustering, we aim to assess the similarity of representations between the two algorithms. In Figure 15 (bottom left), we present the correspondence of the representations with random targets, similar to the approach used in Figure 4 (left), across different training epochs. The darkness of the lines represents the progression of epochs, with lighter shades indicating later epochs. Additionally, in Figure 15 (bottom right), we depict the same correspondence for different intermediate layers, similar to Figure 4 (middle), at the end of the SSL train. Across various layers and epochs, we observe similar behavior between the algorithms. However, it is worth noting that VICReg demonstrates a slightly higher performance in extracting the random targets.

### B.5 Random Target Function Architectures

Neural network architectures possess inherent biases that aid in modeling complex functions. Convolutional networks, for example, implicitly encode biases through their structure, which includes locality and translation invariance. As a result, these networks tend to prioritize local patterns over global features. On the other hand, the ViT architecture takes a different approach by treating images as sequences of patches and utilizing self-attention mechanisms. This design introduces a bias that enables long-range dependencies and global context.

An interesting question arises regarding how the choice of the SSL backbone architecture influences the learned representations. To explore this, we examine the alignment of representations with different types of random targets. Specifically, we conduct experiments similar to those illustrated in Figure 4 (left and middle) but employ different random targets based on the ViT architecture [52].

In Figure 16, we monitor the linear test accuracy of VICReg-trained RES-5-250 in recovering both the ResNet-18 targets and the ViT targets. Figure 16 (left) presents the linear test accuracy at different training epochs (20, 40, 100, 400, 1000) from dark to light resp. and Figure 16 (right) shows the linear test accuracy at different intermediate layers (1-5) from dark to light resp. at the end of SSL

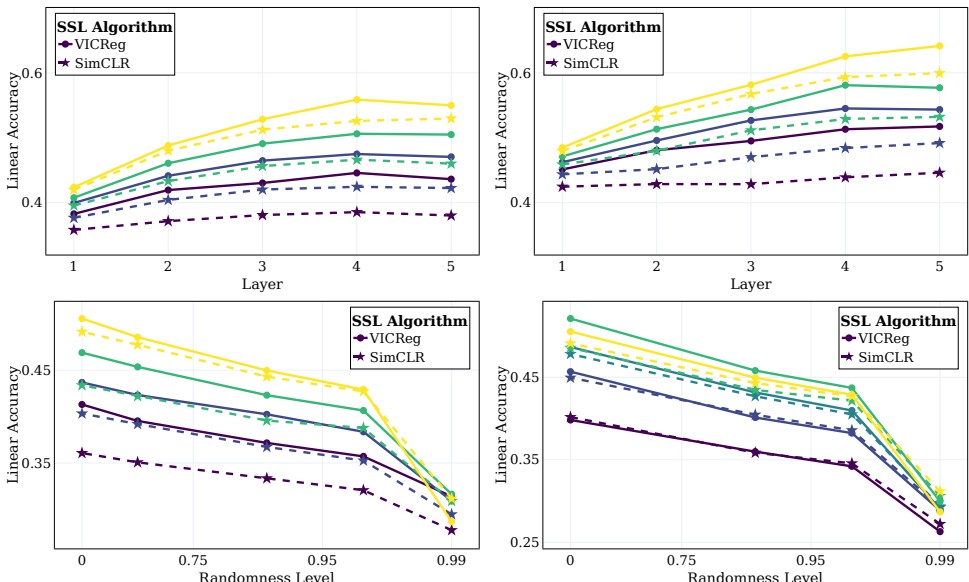

Figure 15: **SimCLR and VICReg have similar performance. (top)** Linear test accuracy in different training epochs, as a function of the intermediate layer, for original classes and superclasses, from left to right resp. **(bottom) (left)** Linear test accuracy in different training epochs (from dark to light) with respect to different randomness levels. **(right)** Linear test accuracy in different intermediate layers, at the end of training with respect to different randomness levels.

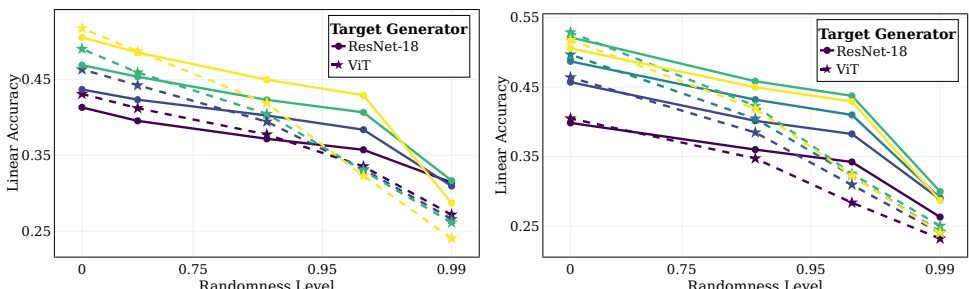

Figure 16: **The implicit bias of the backbone architecture on the learned representations. (left)** Linear test accuracy of an SSL-trained RES-5-250 network for extracting ResNet-18 and ViT random target functions with varying degrees of randomness (x-axis) at different epochs (color-coded from dark to bright). **(right)** Linear test accuracy of an SSL-trained RES-5-250 network for extracting ResNet-18 and ViT random target functions with varying degrees of randomness (x-axis) at different intermediate layers (color-coded from dark to bright).

training. As can be seen, the results with ViT targets are consistent with the claims in Section 5. In other words, the SSL representations exhibit improved alignment during training epochs and across intermediate layers.

However, it is evident that the linear accuracy with respect to the ResNet-18 targets consistently outperforms the accuracy with respect to ViT targets at all training stages, except for the end of the supervised training phase ("0" randomness). This indicates that during the training process, the SSL-trained model acquires representations that are more aligned with those achieved by training convolutional network architectures. This behavior can be attributed to the implicit biases introduced by the backbone, which share similarities with the ones present in the ResNet-18 architecture.

## C   Limitations

While our study has yielded significant findings, it is not without its limitations. Firstly, our experiments were conducted on select datasets, which inherently possess their unique characteristics and biases. As such, applying our analysis to different datasets may potentially give different results. This reflects the inherent variability and diversity of real-world data and the possible influence of dataset-specific factors on our findings. Secondly, our analysis primarily focuses on the vision domain. While we believe that our findings have substantial implications in this area, the generalizability of our findings to other domains, such as natural language processing or audio processing, remains unverified.

## D   Broader Impact

In this paper, our primary objective is to characterize the different properties and aspects of SSL, with the aim of deepening our understanding of the learned representations. SSL has demonstrated its versatility in a wide range of practical downstream applications, including image classification, image segmentation, and various language tasks. While our main focus lies in unraveling the underlying mechanisms of SSL, the insights gained from this research hold the potential to enhance SSL algorithms. Consequently, these improved algorithms could have a significant impact on a diverse set of applications. However, it is crucial to acknowledge the potential risks associated with the misuse of these technologies.

