# OpenReview forum: "Reverse Engineering Self-Supervised Learning"
_NeurIPS.cc/2023/Conference — NeurIPS 2023 poster_

### Official Review · Reviewer_b2bW · 2023-06-29

**Soundness:** 3 good
**Presentation:** 3 good
**Contribution:** 3 good
**Rating:** 6
**Confidence:** 5

**Summary:**

This work presents a study of the structure of SSL representations with respect to sample labels (image identity), to class labels and to super-class labels. To that end, an analysis of the clustering properties of the embedding space according to such labels is performed. Additionally, probing linear layers are trained on top of the SSL representations to assess discriminative performance for the different label categories.

VicReg on CIFAR-100 is evaluated in the experimental results section. The main results show that representations indeed cluster according to multiple hierarchies of labels (sample, class, super-class) and that such clustering is mostly related to the regularization term in VicReg.

**Strengths:**

### Clarity

* The language used is clear.
* The paper organization seems natural and easy to follow.

### Significance

* Investigating the intrinsic properties of SSL representations is of interest to the community. However, a ore throrough analysis would be needed (see Weaknesses).

**Weaknesses:**

### Originality

* The analyses provided in this work have already been (at least partially) explored. This reduces the originality of the work. See some references hereafter, not being an exhaustive list:
  * To the best of my knowledge, the fact that SSL representations cluster according to class labels has already been explored in the past, see (Vasequi, 2021) for example. Although such work focused on label noise resiliency, they show clear evidence of clustering in the t-SNE plots provided.
    - _Zahra Vaseqi, Ibtihel Amara and Samrudhdhi Rangrej, Label Noise Resiliency with Self-supervised Representations, NeurIPS 2021_.

  * (Grigg, 2021) also showed that linear probing performance increases with depth, in that case with SimCLR representations.
    - _Tom George Grigg, Dan Busbridge, Jason Ramapuram and Russ Webb, Do Self-Supervised and Supervised Methods Learn Similar Visual Representations?, NeurIPS 2021_.

### Quality

* The experimental results in the main paper only cover 1 SSL algorithm (VicReg) and 1 dataset (CIFAR-100). Such limited evaluation setup does not reflect the main claim of the paper (reverse-engineering SSL), being rather a study of VicReg properties on a specific dataset. It is yet to be seen whether the results obtained with VicReg transfer to:
  * The myriad of SSL algorithms previously published (i.e. SimCLR, SimSiam, BYOL, SwAV, Barlow Twins, etc.), some of them much more widely used in the community than VicReg.
  * Different datasets with different properties, specially different inter-class similarities which can strongly affect clustering. Such analysis would give the reader intuition about how different data sources can be expected to work in an SSL framework.

### Clarity

* The fact that the authors refer to SSL-wide conclusions while using conclusions on VicReg+CIFAR100 induces the reader to misunderstanding. I would suggest the authors to either (i) provide a thorough analysis over representative SSL methods and datasets or (ii) refer to VicReg properties instead of general SSL.

**Questions:**

* The claim _"Contrary to popular belief, the invariance loss does not significantly improve during the training process"_ (L257) is not rigorously proven. Have the authors verified this property on a suite of SSL methods, to be able to state it as formulated in the manuscript? I would argue that this is indeed a property of VicReg in the setting of the paper. However, I am doubtful about this finding being a generic SSL property. For that, a much larger study of SSL methods, with different loss term weighings, etc. would be required.

* I am curious about how much the super-class clustering is different from a random super-class assignment (in groups of 5 classes, similarly to the experimental setup already in the paper). Is this related to the label-noise experiment, or would this random grouping yield different results?

-----
Minor comments:
* Fig 2 right is not referenced.

-----
Main reasons behind score:

* Lack of comparison with other SSL papers or datasets. The representation properties analyzed are only based on VicReg, which strongly reduces the significance of this work and contradicts the main contributions (general SSL).
* Some observations explored in this work are not original, and had been explored in previous works.

-----
Update after rebuttal:

The authors did a great work in the rebuttal, largely improving the experimental sections and thus addressing most of my concerns. In order to avoid ambiguity, I raise my score to 5.

The authors also included ImageNet results that support the paper claims, making the paper much stronger as large scale data is also part of the evaluation. Having 4+ methods on several datasets makes the claims of the paper more sound and provide interesting insights to the community. Therefore, after the rebuttal conversations, I raise my score to 6.

**Limitations:**

No concerns.

---

> ### Author Rebuttal · Authors · 2023-08-09
>
> Dear Reviewer b2bW,
>
> Thank you very much for your thorough review and feedback on our paper.
>
> We value your constructive comments and would like to address the weaknesses and questions you raised:
>
> # Weaknesses:
>
> ## Originality:
>
> We appreciate the criticism that has been raised while also maintaining our confidence in the originality of our paper, which introduces several innovative concepts that have yet to be explored in previous research. Specifically, our study delves into SSL algorithms from the unique perspectives of Neural Collapse[6], Randomness, and Information Compression. We extend our gratitude to the reviewer for pointing out the two relevant references, which we will duly address in the revised version of our manuscript. Let us explicitly highlight the distinctions between their contributions and our own.
>
> * **(Vasequi, 2021)**  The focus of this paper is on investigating the robustness of SSL methods with respect to label noise. While a t-SNE visualization of representation from a single pair of networks is briefly presented, the paper does not quantify, study the intermediate relations, and view the SSL algorithms from a geometrical perspective. The clustering of SSL is briefly mentioned, using a visualization - while in our work, we aim to quantify and asses the intrinsic geometries, along with a visualization in Figure 1.
>
> * **(Grigg, 2021)** Thank you for the provided reference. Even though the paper analyzed the linear probing of SimCLR on ResNet50, it does not analyze the clustering aspect with respect to the dataset the SSL was trained on. Rather it studies the similarity between representations of SSL with those of Supervised Learning. This work does not study the geometry formed in different SSL algorithms alone. Rather it compares the representations with those of the supervised setting. Moreover, our paper stands out for its original analysis through the lenses of Neural Collapse, Randomness, and Information Compression, a perspective that sets our work apart.
>
> ## Quality:
>
> We're sorry for the possible confusion in our manuscript. We want to emphasize that even in our original manuscript, there are three datasets (CIFAR100, CIFAR10, and FOOD101) and two SSL algorithms (VICReg and SimCLR - see appendix for more details). Prompted by the reviewer's feedback, we've expanded our experiments and analysis.
> - **Datasets** - Additional to CIFAR100, CIFAR10, and FOOD101, we included results on Tiny-ImageNet [1], Aircrafts [2], Oxford Pets [3], and Oxford Flowers [4]. These datasets are much more diverse and contain different features, different numbers of training examples, and different numbers of classes.
> - **SSL Methods** - To further enrich our study, we've added to VICReg and SimCLR, the clustering-based SSL method, SWaV, and Barlow Twins.
> - **Integration with Recent Architectures as Backbones** - Additional to RES-L-W of different widths and depths, we included Vision Transformer (VIT) [5] as an additional backbone.
>
> As you can see in the attached page, these new models, SSL methods, and architectures exhibit similar behavior to the current experiments. These additions strengthen our paper by providing a more comprehensive and robust analysis.
>
> ## Clarity:
>
> As mentioned above, we extensively extended our analysis for different datasets (CIFAR100, CIFAR10, FOOD101, Tiny-ImageNet, Aircrafts, Oxford Pets, and Oxford Flowers), SSL algorithms (VICReg, SimCLR, SwAV, BarlowTwins), and backbones (RES-5-250, RES-10-250, RES-5-1000, ViT).
>
> # Questions
> * **Invariance loss during training:** We thank the reviewer for highlighting the gap in our manuscript. Given the reviewer's guidance, we present the decomposition of the loss terms to invariance and regularization terms for SimCLR in the attached file. Figure 3 (left) shows that the observed behavior is similar to VICReg. In the revision of our paper, we will add this analysis to our existing SimCLR section.
> * We thank the reviewer for the interesting question! In Figure 4 (Left) of the attached pdf, we've added the superclass clustering in different layers and randomness levels. Figure 4 (Left) in the main text (of the semantic labels) is similar to that of the superclasses. It is additionally apparent that the superclass clustering improves with respect to decreasing randomness.
>
> # Minor Comments
> Figure 2 (right) is mentioned in L332 regarding the clustering with respect to different hierarchies. We will further reference it for brevity.
>
>
> Thank you again for your thoughtful review. We made an effort to address your feedback, including multiple experiments and paper edits, and we would greatly appreciate it if you would consider raising your score in light of our response.  Please let us know if you have additional questions we can address.
>
> # References
> [1] Y. Le et al., "Tiny ImageNet Visual Recognition Challenge", 2015.
>
> [2] S. Maji et al., "Fine-Grained Visual Classification of Aircraft", 2013.
>
> [3] Zhang, H. et al. "0/1 Deep Neural Networks via Block Coordinate Descent", 2022.
>
> [4] M.-E. Nilsback et al., "Automated Flower Classification over a Large Number of Classes",  2008.
>
> [5] A. Dosovitskiy et al., "An Image is Worth 16x16 Words: Transformers for Image Recognition at Scale" , 2021.
>
> [6] V. Papyan, et al., "Prevalence of neural collapse during the terminal phase of deep learning training," 2020.

---

> > ### Comment · Reviewer_b2bW · 2023-08-15
> > **Answer to rebuttal**
> >
> > I would like to thank the authors for addressing the questions raised during the review with additional experiments.
> >
> > The results presented, including SimCLR and SwAV, as well as more datasets and backbones highly improve the soundness of this work. I believe now it more accurately reflects the spirit of the work: analysing SSL beyond a single method.
> >
> > It is still an open question how the findings of this work would transfer to large scale datasets (for example, Flowers has only  ~1000 images in the training set). However, I think the authors did a great effort in running several extra experiments in short time. It would be great to have some result on ImageNet or larger datasets, but I believe the paper is interesting already as is.
> >
> > I specifically thank the reviewers for addressing the randomness levels question.
> >
> > Overall, I believe the manuscript has improved substantially and I am willing to increase my score to 5.

---

> > > ### Author Response · Authors · 2023-08-18
> > > **Response to Reviewer b2bW’s Second Comment**
> > >
> > > Dear Reviewer b2bW,
> > >
> > > Thank you for your feedback, for engaging with us, for your willingness to revise your initial assessment, and for recommending acceptance of our submission!
> > >
> > > In response to **your comment regarding large-scale datasets**, we have taken this concern seriously and have now **conducted further experiments using the ImageNet dataset** with ResNet50 with **several SSL methods including VICReg, MoCoV2, SwAV and BarlowTwins**.  We are encouraged to observe trends similar to those identified in our previous analyses on other datasets. To share some specific numbers, The NCC of the last layer for different SSL methods on the ImageNet dataset are as follows:
> > >
> > > | **Random Init**         | **SwAV**       | **VICReg**     | **BarlowTwins** | **MoCoV2**     |
> > > |------------------|----------------|----------------|-----------------|----------------|
> > > | 0.03522386818    | 0.2700982024   | 0.4742010652   | 0.3506574567    | 0.4009445739   |
> > >
> > > As evidenced by the results, the SSL methods contribute significantly to the clustering process, demonstrating strong clustering behavior as training progresses. **This aligns with our broader observations and further supports our hypothesis and contributions.**
> > >
> > > Unfortunately, due to the limitations of the open review system, we are unable to share visual figures of these results at this point. We understand the impact that figures can have in portraying results, and we are committed to including these in the final version of the paper, should it be accepted.
> > >
> > > Lastly, we are thankful for your specific acknowledgment regarding the randomness levels question. We endeavored to address this thoroughly, as we understand the potential implications it has for the robustness and generalizability of our findings.
> > >
> > >
> > > We specifically conducted the experiments that you requested and hope that you can take these efforts into account in your evaluation. **We kindly ask that you consider these contributions when reassessing our work, and we hope that they warrant an increase in our score.** Thank you once again for your kind words and for the invaluable role you have played in strengthening our manuscript.

---

> > > > ### Comment · Reviewer_b2bW · 2023-08-20
> > > > **On ImageNet experiments**
> > > >
> > > > First of all, apologies for the late answer and the uncertainty this might cause during the reviewing process.
> > > >
> > > > Thanks for running the ImageNet experiments. In my opinion, these experiments were required for acceptance at a conference like NeurIPS, given the scope of the work (understanding aspects of SSL). SSL is valuable at large scale, so findings on small datasets might or might not be meaninful after all. Having ImageNet experiments and showing that the results are still in line with the more toyish datasets is great!
> > > >
> > > > I find the fact that the NCC accuracy reaches ~0.4 for ImageNet very interesting. It is higher than for much smaller datasets, which gives the intuition that there are 2 aspects playing a role here: (1) intuitively more classes should make the NCC go down, since the task becomes harder, however (2) the fact that there are more images largely improve the representation expressivity (better features) making the NCC go up. I wonder if the authors agree with this thought. If they do, this could be measured by computing the NCC of a backbone trained on Flowers and tested on Flowers, and the NCC of a backbone trained on ImageNet and tested on Flowers (one could also finetune on Flowers before testing). My intuition is that the better features learnt on ImageNet will also give better NCC on Flowers. I want to make sure the authors understand that this question is purely out of curiosity, and not a request of any sort.
> > > >
> > > > Overall, having several small-medium scale datasets together with ImageNet, on 4-5 different SSL methods makes this work much more valuable than the original submission. I believe that including ImageNet (taking the authors word to include them in the final manuscript) is a reason enough to raise the score to 6.

---

### Official Review · Reviewer_MBnG · 2023-07-06

**Soundness:** 3 good
**Presentation:** 3 good
**Contribution:** 3 good
**Rating:** 7
**Confidence:** 5

**Summary:**

This paper provides an empirical study on self-supervised learning (SSL) in terms of clustering. Specifically, the authors demonstrate 1) clustering process of SSL methods, 2) semantic learning of SSL methods based on random labels generated by various models, and 3) class hierarchies and intermediate layers of SSL methods.

**Strengths:**

- The presentation of this paper is really good: overally well-written, and easy-to-follow.

- The message of this paper is clear, sound, and consistent to me: self-supervised learning works as a clustering algorithm of feature space, and various empirical observations support it well.

- The experiments are conducted in depth with convincing (but seems narrow) setups.

**Weaknesses:**

The experiments seems not extensive in terms of dataset and self-supervised algorithms as follows:

- Many self-supervised learning methods are evaluated using ImageNet dataset, and this paper lacks experimental results on ImageNet dataset. This is not critical for accepting this paper, but additional experimental results on ImageNet dataset would improve the message of this paper.

- This paper mainly focuses on one self-supervised learning, VICReg, which may not represent all self-supervised learning methods. This is also not that critical.

**Questions:**

I understand that one of "contrastive" SSL methods works as clustering algorithm, but wonder whether it can be applied to "cluestering-based" SSL (e.g., [SWaV](https://arxiv.org/pdf/2006.09882.pdf)) and "auto-regressive" SSL (e.g., [MAE](https://arxiv.org/pdf/2111.06377.pdf)). I believe that this is not in the scope of this paper, but it would be better to discuss it for the future research direction.

**Limitations:**

The authors provide limitations of this paper in Appendix.

---

> ### Author Rebuttal · Authors · 2023-08-09
>
> Dear Reviewer MBnG,
>
> Thank you very much for your thorough review and positive feedback on our paper. We are pleased you found our work well-written, clear, and consistent. Your encouraging words on our study's presentation, soundness, and contribution are greatly appreciated.
>
> We value your constructive comments and would like to address the weaknesses and questions you raised:
>
> ## Weaknesses:
>
> 1. **Additional Datasets, SSL Methods, and Architectures**: First, we would like to note that besides our results on VICReg with the ResNet backbone on CIFAR100, we have additional results in our appendix using the SimCLR SSL method and the Food dataset. We focused on these datasets and methods due to their simplicity on the one hand and their common use in the literature on the other hand. However, we agree that the analysis would be enriched by considering more datasets, SSL models, and architectures. Prompted by your feedback, we have added a few additional experiments. We ran our analysis on the following:
> - **Datasets** - Additional to CIFAR100, CIFAR10, and FOOD101, we included results on Tiny-ImageNet [1], Aircrafts [2], Oxford Pets [3], and Oxford Flowers [4]. These datasets are much more diverse and contain different features, different numbers of training examples, and different numbers of classes.
> - **SSL Methods** - To further enrich our study, we've added to VICReg and SimCLR, the clustering-based SSL method, SWaV, and Barlow Twins.
> - **Integration with Recent Architectures as Backbones** - Additional to RES-L-W of different widths and depths, we included Vision Transformer (VIT) [5] as an additional backbone
> As you can see in the attached page, these new models, SSL methods, and architectures exhibit similar behavior to the current experiments. These additions strengthen our paper by providing a more comprehensive and robust analysis.
>
>
> As you can see in the attached page, these new models, SSL methods, and architectures exhibit similar behavior to the current experiments. These additions strengthen our paper by providing a more comprehensive and robust analysis.
>
> ## Questions:
>
>
> **Applicability to Other SSL Methods**: As mentioned above, based on your suggestion, we ran our analysis on SWaV (see Figure 3 in the attached page). As you can observe, the behavior is even clearer and more pronounced compared to contrastive (VICReg) and non-contrastive (SimCLR) methods. This behavior makes sense, as the SWaV loss is designed specifically for clustering. Therefore, even though clustering is not conducted explicitly with specific labels, it appears much more significant compared to methods that do not explicitly enforce clustering and only look at pairs of images.
>
> Regarding the "auto-regressive" SSL, since only recently, MAE methods succeeded in achieving state-of-the-art performance in vision datasets [6], we didn't investigate them. However, our intuition is that running our analysis on these methods will give results similar to what we obtain when we look at the hidden space of unsupervised methods [7]. Namely, the features are based on the directions with the highest variance in the input (similar to nonlinear PCA). It would be interesting to explore the differences in representation between SSL and unsupervised learning, as, in principle, we can compare features designed for generation (unsupervised learning) and those designed for the classification of downstream tasks (SSL).
>
>
>
> Thank you again for your thoughtful review. We made an effort to address your feedback including multiple experiments and paper edits, and we would greatly appreciate it if you would consider raising your score in light of our response. Please let us know if you have additional questions we can address.
>
>
>
>
>
> ## References
> [1] Y. Le et al., "Tiny ImageNet Visual Recognition Challenge", 2015.
>
> [2] S. Maji et al., "Fine-Grained Visual Classification of Aircraft",  2013.
>
> [3] Zhang, H. et al. "0/1 Deep Neural Networks via Block Coordinate Descent", 2022.
>
> [4] M.-E. Nilsback et al., "Automated Flower Classification over a Large Number of Classes", 2008.
>
> [5] A. Dosovitskiy et al., "An Image is Worth 16x16 Words: Transformers for Image Recognition at Scale" , 2021.
>
> [6] K. He et al., "Masked Autoencoders Are Scalable Vision Learners", 2022.
>
> [7] I. Higgins et al., "beta-VAE: Learning Basic Visual Concepts with a Constrained Variational Framework",  2017.

---

> > ### Author Response · Authors · 2023-08-20
> > **Additional Experiments on ImageNet**
> >
> > Dear Reviewer MBnG,
> >
> > The discussion period ends in 24 hours.
> >
> > Thank you once again for your thoughtful and comprehensive feedback. We have made a significant effort to address the questions and concerns raised in your review, including additional datasets (more than five different datasets), SSL methods (more than five methods), and backbone architectures (RES-L-W, Vision Transformers, and ResNets in different sizes). We would like to point out that we have even added analysis on the **ImageNet dataset for various SSL methods** following your feedback.
> >
> > We have made every effort to address your concerns in our revised manuscript through extensive experiments and clarifications. We kindly request that you consider these in your final assessment and consider raising your score in your final response.
> >
> > Please do not hesitate to let us know if there are additional points we can address.

---

### Official Review · Reviewer_YeWL · 2023-07-06

**Soundness:** 2 fair
**Presentation:** 3 good
**Contribution:** 2 fair
**Rating:** 5
**Confidence:** 3

**Summary:**

This paper focuses on an empirical analysis of how self-supervised learning (SSL) works. While training artificial neural networks for classification under supervision has been extensively explored, the authors propose to shed more light on SSL process using VicReg + ResNet-L-H and a series of experiments on the CIFAR-100 dataset. This work presents that during the training process, we can observe improvements in clustering for the original classes and in super-classes, that presents some emerging properties of hierarchical clustering during SSL training. What is important, this happens while the expected instance-level clustering stops improving. Although the authors do not provide an answer as to why this is the case, they attribute this phenomenon to the regularization term in the SSL loss function (what is presented in one of the experiments). In work we can find few more analysis.

**Strengths:**

S1. This paper tackles an important problem of understanding the representations built by SSL and asks important questions, e.g. how the clustering evolves during the training on multiple levels (instance, class semantic, high level semantic) and to what this phenomenon can be attributed.

S2. The paper shows how the SSL methods gain clustering properties across training procedure (Fig. 2) and across layers (Fig. 5). While this property was intuitively known, this paper is the first to measure this property.

**Weaknesses:**

W1. The paper draws conclusions based on experiments that are made using only one SSL method, one dataset (CIFAR100 - which is small scale) and one type of neural network architecture (RES-L-H which is not widely used).

W2. The contributions regarding Randomness impact do not sound solid. The randomness experiment is not clear (see questions sections) and it is hard to know what it tries to show. By this experiment, the authors want to show that “We argue that SSL-trained representation functions are highly correlated with semantic classes.”. Doesn't the S1 already show it?
Similarly, does not the fact that the classes are clustered would not imply that superclasses are clustered as well? I think it is a nice thing to show, but currently it is exposed as one of the main paper contributions. Presenting that some (super) emerges before the original - would be more insightful

W3. This work is a series of empirical analysis, with some insightful information. However, the reading flow could be improved for that.

**Questions:**

1. One of the insightful outcome of this work is that the clustering property can be attributed to regularization term. But this was not analyzed. Why not increase the term for that component and check the behavior?

1. Looking at the figure 3 (right). The SSL method seems to work significantly worse than the SL method (41 vs 61% accuracy). This looks like a massive difference, maybe there is something wrong with the training run?

1. Can we see figure 2 without normalized accuracies? I think the default option would be to show absolute values, so I wonder what we gain by showing normalized accuracies. One culprit is that in your “Sample” experiment you have 500 classes and in “original” 100 classes, but I would still like to see absolute values.

1. The randomness experiment is not clear to me. The paper says: „We train a neural network classifier on the same dataset for classification and use its target predictions at different epochs as targets with varying degrees of randomness”, which I found hard to understand. Are the labels simply random at training time as in [1]?

Also: “We normalize the degree of randomness between 0 (non-random, end of training) and 1 (random, initialization)” - how do you do this? Does 0.9 degree of randomness correspond to 0.9 fraction of total training epochs? Or does it correspond to 0.9 of final performance of the network?
Finally, looking at the fig. 4. Is there something surprising in the plot? (Increased randomness = reduced accuracy).

1. I think there is something wrong with the following sentence: L.210: “Notably, the SSL-trained representation exhibits neural collapse at a sample level (with an NCC train accuracy near 1.0), yet the clustering with respect to the semantic classes is also significant (≈ 0.41 on the original targets).“ Maybe it is about Fig. 3 (right) but the NCC train accuracy for the SSL method is above 0.4 (not “near 1.0”)?
Also, it is not clear what is the message that this figure shows as i) SL and SSL have totally different accuracies so they are not comparable, ii) the fact that we get worse results on validation is also expected.

1. “As shown in Figure 5, we observe that the clustering and separation ability of each layer improves as we move deeper into the network” - it is not true. For classification of original semantic classes the linear accuracy actually drops at the last layer (or several last layers in case of not fully trained networks).

1. Why decision to show only one dataset and type of architecture in the main paper? I know that in the appendix there is a food dataset and different variation of resnet-l-h. However, why not going for imagenet (or subset)?

1. Have you considered checking methods like SupCon [2] for a comparison?

[1] Zhang, Chiyuan, et al. "Understanding deep learning (still) requires rethinking generalization." Communications of the ACM 64.3 (2021): 107-115.
[2] Khosla, Prannay, et al. "Supervised contrastive learning." Advances in neural information processing systems 33 (2020): 18661-18673.

**Limitations:**

Limitations are not discussed.

---

> ### Author Rebuttal · Authors · 2023-08-09
>
> We thank the reviewer for the valuable feedback. We've taken your insights into account and have expanded our analyses and clarifications as detailed below.
> # Weaknesses
> ## W1:
> While this work is an initial study into several geometrical phenomena in SSL, we've attempted to incorporate many networks and architectures. Most works in SSL [6-9] use a single backbone (e.g., RN50), and vary the loss. We've chosen the RES-L-H as it is a variant of the ResNet architecture with the same dim. in every block so that the intermediate performance can be compared. In the appendix, we've also shown results using CIFAR10 and FOOD101 datasets. Following the reviews, we strengthened the empirical results:
> - **Datasets** - Additional to CIFAR100, CIFAR10, and FOOD101, we included results on Tiny-ImageNet [1], Aircrafts [2], Oxford Pets [3], and Oxford Flowers [4]. These datasets are much more diverse and contain different features, different numbers of training examples, and different numbers of classes.
> - **SSL Methods** - To further enrich our study, we've added to VICReg and SimCLR, the clustering-based SSL method, SWaV, and Barlow Twins.
> - **Integration with Recent Architectures as Backbones** - Additional to RES-L-W of different widths and depths, we included Vision Transformer (VIT) [5] as an additional backbone.
>
> As you can see in the attached page, these new models, SSL methods, and architectures exhibit similar behavior to the current experiments. These additions strengthen our paper by providing a more comprehensive and robust analysis.
>
> ## W2:
> While we agree that the clustering phenomena point to agreement with semantic classes, the randomness experiment sheds light on how this clustering occurs. Namely, we claim that not only does the data cluster with respect to semantic labels, this clustering increasingly matches "less random" repr. In our work, "less random" means more aligned with SL on the same task. We're showing that not only does the clustering happen, but it becomes more aligned with the supervised training process on the same dataset.
>
> We believe that clustering w.r.t superclasses is not trivial. The fact that classes are clustered does not imply that the superclasses are clustered. Consider two super classes with classes {1,2} and classes {3,4}. We can think of samples lying on a line. The samples in class 1 concentrate around x=1, the samples of class 2 concentrate around x=4, the samples of class 3 concentrate around  x=2, and the samples of class 4 concentrate around x=3. The samples themselves are clustered, but not with respect to their superclasses.
>
> ## W3:
> We thank the reviewer for this feedback and will attempt to improve the flow in the revised version.
>
> # Questions
> 1. Thank you for this question. This experiment was shown in the Appendix (see Fig 10). We show that with a low regularization term, the linear accuracy greatly decreases. However, with a very large regularization term, the linear performance remains near optimal.
> 2. We believe the gap is not due to suboptimal training but rather to applying SSL. When training the model with SSL, it does not have access to labeled data, and therefore, we cannot expect a linear probe to perform as well as SL without fine-tuning the pre-trained model.
> 3. The Appendix provides this analysis (See Fig 6 in).
> 4. We will attempt to further describe the process in the revised version. The randomness presented (unlike the reference) discusses the degree of agreement with a supervised classifier on the same dataset. This means that 1.0 randomness is the predictions made by a network at initialization, and 0.0 randomness is the predictions made by a trained supervised classifier.
> 5. The randomness corresponds to 0.9 fraction of total training epochs. While the result is somewhat intuitive, it shows a strong connection between the representations learned in SSL and those of a supervised classifier on the same task. This shows that the SSL incrementally learns representations more inline with supervised training. Further, as is shown in Appendix B.5, different networks used for the supervised prediction behave differently with respect to SSL.
> 6. We apologize if this sentence is not clear. It refers to Fig 2 (Left), yet it is not shown in the figure as it is normalized. Looking at Fig 6 (Middle) in the Appendix, we see that the sample-level clustering is near 1.0, and the original-label-level clustering is 0.41.
> 7. Thank you for this comment. A more subtle statement may say that there is a general monotonic clustering trend in the intermediate layers, while some saturation (and even lower NCC) is in the final layers of the network.
> 8. We decided to focus on a single architecture during the main paper to show the myriad of results in our work exhaustively.
>
> Thank you again for your thoughtful review. We made an effort to address your feedback, including multiple experiments and paper edits, and we would greatly appreciate it if you would consider raising your score in light of our response.  Please let us know if you have additional questions we can address.
>
>  ## References
> [1] Y. Le et al., "Tiny ImageNet Visual Recognition Challenge", 2015.
>
> [2] S. Maji et al., "Fine-Grained Visual Classification of Aircraft", 2013.
>
> [3] Zhang, H. et al. "0/1 Deep Neural Networks via Block Coordinate Descent", 2022
>
> [4] M.-E. Nilsback et al., "Automated Flower Classification over a Large Number of Classes", 2008
>
> [5] A. Dosovitskiy et al., "An Image is Worth 16x16 Words: Transformers for Image Recognition at Scale",  2021
>
> [6] A. Bardes, et al., "VICReg: Variance-Invariance-Covariance Regularization for Self-Supervised Learning", 2022
>
> [7] T. Chen, et al., "A Simple Framework for Contrastive Learning of Visual Representations", 2020
>
> [8] J.-B. Grill, et al., "Bootstrap Your Own Latent a New Approach to Self-Supervised Learning", 2020
>
> [9] M. Caron, et al., "Unsupervised Learning of Visual Features by Contrasting Cluster Assignments", 2020

---

> > ### Author Response · Authors · 2023-08-19
> >
> > Dear Reviewer YeWL,
> >
> > Thank you for your thoughtful review. We greatly appreciate your time and effort in evaluating our manuscript. We have worked diligently to address your concerns and hope that our revisions demonstrate the novelty and significance of our results.
> >
> > Specifically, we've incorporated your comments, and the proposed experiments are in the attached pdf or the appendix. Concerning **W1**, we've added additional experiments on different datasets, SSL methods, and backbones (including RES-L-W of different widths and depths, ViT, and ResNets). Concerning **W2**, we've addressed the questions about randomness, why it differs from label noise in the literature,  and how it contributes to our work. Regarding **W3**, we will improve the writing flow to clarify the research presented in this work. We would be glad to address any additional concerns you have.
> >
> > In addition to the experiments outlined in our previous rebuttal, which include the evaluation of various models, SSL methods, and architectures, we have considered your feedback and conducted several additional extensive experiments. Notably, these new experiments were performed on the **ImageNet** dataset, which contains 1,281,167 images for training and 50,000 images for validation.
> >
> > In response to your insightful suggestions, we have conducted further experiments using the ImageNet dataset with ResNet50 and several SSL methods, including **VICReg, MoCoV2, SwAV, and BarlowTwins**. We are encouraged to observe trends similar to those identified in our previous analyses on other datasets. For example, the NCC of the last layer for different SSL methods on the ImageNet dataset is as follows (for full details, please refer to our general response):
> >
> > | **Random Init**         | **SwAV**       | **VICReg**     | **BarlowTwins** | **MoCoV2**     |
> > |------------------|----------------|----------------|-----------------|----------------|
> > | 0.03522386818    | 0.2700982024   | 0.4742010652   | 0.3506574567    | 0.4009445739   |
> >
> > These results show that the SSL methods contribute significantly to the clustering process, demonstrating strong clustering behavior as training progresses. These findings align with our broader observations and provide robust support for our hypotheses and contributions.
> >
> > Unfortunately, due to the open review system's limitations, we cannot share visual representations of these results at this point. We fully understand the impact figures can have in portraying results, and we are committed to including these in the final version of the paper, should it be accepted.
> >
> > We are dedicated to integrating these new results into the camera-ready version of the paper and presenting them clearly through figures. This additional analysis significantly strengthens our work by demonstrating our findings for large datasets.
> >
> > We kindly ask that you consider these contributions when reassessing our work, and we hope that they warrant an increase in our score.
> > Thank you immensely for your constructive and insightful feedback. We appreciate your consideration of our paper and welcome further suggestions.

---

> > > ### Author Response · Authors · 2023-08-20
> > > **Request for Engagement**
> > >
> > > Dear Reviewer YeWL,
> > >
> > > The discussion period ends in 24 hours.
> > >
> > > We have made a significant effort to address the questions and concerns raised in your review, including additional datasets (more than five different datasets), SSL methods (more than five methods), and backbone architectures (RES-L-W, Vision Transformers, and ResNets in different sizes). We would like to point out that we have even added analysis on the **ImageNet dataset for various SSL methods**, as mentioned in the general comment.
> > >
> > > We have made every effort to address your concerns in our revised manuscript through extensive experiments and clarifications. We kindly request that you consider these in your final assessment and consider raising your score in your final response.
> > >
> > > Please do not hesitate to let us know if there are additional points we can address.

---

> > > > ### Comment · Reviewer_YeWL · 2023-08-21
> > > > **I appreciate the effort, the submission improved, maybe not fully support the approach of making this research**
> > > >
> > > > This work raised a lot of questions, prompted some discussions, and undoubtedly required a significant effort from the authors during the rebuttal stage. I've read other reviewers' comments and your answers. Thank you for addressing my issues and answering my questions. I am willing to raise my score.
> > > >
> > > > Just a reminder about the rules:
> > > > (NeurIPS 2023 Conference Authors: Discussion Period) rules:
> > > > "Please do not post messages that urge reviewers to respond. The AC and PCs will send these"

---

### Official Review · Reviewer_7nAJ · 2023-07-07

**Soundness:** 3 good
**Presentation:** 3 good
**Contribution:** 2 fair
**Rating:** 5
**Confidence:** 3

**Summary:**

To reveal the underlying mechanism of self-supervised learning methods, this paper mainly investigate their clustering characteristics. Empirical analyses show that these methods cluster samples based on their semantic classes. Additionally, the use of regularization terms, introduced to prevent representation collapse, proves beneficial even after the clustering process.

**Strengths:**

1. Understanding the mechanism of self-supervised learning is one of the most important goals acknowledged by the machine learning community.
2. This paper provides not only qualitative but also quantitative results.
3. This paper is well-written and well-organized. Notably, the introduction section effectively summarizes the key takeaways

**Weaknesses:**

1. While the findings are interesting, I'm not fully convinced of their importance. A comparison with other pre-training methods or the proposal of a novel self-supervised learning method based on the conclusions drawn would emphasize the significance of the paper's findings.
2. The scope of the setup is limited:
    - The experiments solely utilize RES-L-H architectures.
    - The primary downstream task focuses on image classification using CIFAR-100. This not only neglects dense prediction tasks but also relies on a small dataset compared with ImageNet.
    - Many experiments mainly focuses on linear probing accuracy rather than fine-tuning performance. However, a high linear probing accuracy does not necessarily indicate a high fine-tuning accuracy [a]. In practice, fine-tuning accuracy is sometimes more crucial than linear probing accuracy.
    - The title and the term “self-supervised learning” in this paper are potentially misleading, since this paper mainly focuses on contrastive and non-contrastive learning, rather than on masked image modeling. Moreover, it seems that some claims are only valid for contrastive or non-contrastive learning.

[a] Park, Namuk, et al. "What Do Self-Supervised Vision Transformers Learn?." ICLR (2023).

**Questions:**

One of my major concerns is about the significance of the findings, since the paper primarily focuses on describing the properties of contrastive and non-contrastive learning methods.

**Limitations:**

I cannot find any ethical concerns. For technical limitations, see the Weaknesses section above.

---

> ### Author Rebuttal · Authors · 2023-08-09
>
> We thank the reviewer for the valuable feedback. We've taken your insights into account and have expanded our analyses and clarifications as detailed below.
>
> # Weaknesses:
> ## W1:
> While we agree that deriving a novel SSL method based on these findings could be a compelling avenue for future exploration, we believe our current work lays the foundational understanding of the principal mechanisms through which SSL methods learn semantic labels. This parallels the work shown in [1], where the phenomenon of Neural Collapse was first introduced and subsequently utilized to improve generalization and performance in [2,3]. Shedding light on the connection between the learning patterns of supervised algorithms with those of SSL, as presented in this paper, is a significant leap toward improving SSL. For example, in L347, "Our study extends..". This outcome could guide decisions regarding selecting intermediate layers when leveraging representations from an SSL algorithm. While this is outside the scope of our current work, it certainly presents a promising avenue for future research!
>
> ## W2:
>
> In this work, our primary objective was to study the relationship between supervised classification and SSL. Our findings underscore that the Neural Collapse framework [1] and linear probing—across diverse architectures, datasets, SSL algorithms, and levels of randomness — are intrinsically linked to the supervised context. We have demonstrated that SSL algorithms can deduce semantic labels during training in a manner reminiscent of the supervised setting. Although we did not incorporate ImageNet in our analyses due to a lack of computational resources, we have attempted to address the reviewer's concerns by extending our studies to several other datasets, including Tiny-ImageNet [8], Aircrafts [9], Oxford Pets [10], and Oxford Flowers [11]. Future research may study how clustering in SSL algorithms impacts different downstream applications, such as Object Detection and Segmentation.
>
> We agree that linear probing might not fully capture the performance realized during fine-tuning. However, the primary focus of this paper was to analyze the representations acquired during SSL within the same dataset. The influence of clustering degree on performance during fine-tuning indeed presents an intriguing avenue for exploration. In most cases, linear probing serves as a measure to quantify the representation by decoupling it from downstream fine-tuning [4-7]. Our specific interest lies in understanding how SSL algorithms cluster data in relation to previously unseen labels and how this process diverges from the findings in the supervised case [1]. Given this objective, it's more pertinent for our study to examine the representations prior to any fine-tuning.
>
> We acknowledge that the primary focus of our study revolves around contrastive and non-contrastive methods, and we will ensure this is clearly communicated in the paper. It's important to highlight that most SSL algorithms [4-7], could be categorized under our framework. Investigating how MAE-based methods fit into our framework is an interesting avenue for further exploration.
>
> To further extend our analysis, we've added:
>
> - **Datasets** - Additional to CIFAR100, CIFAR10, and FOOD101, we included results on Tiny-ImageNet [8], Aircrafts [9], Oxford Pets [10], and Oxford Flowers [11]. These datasets are much more diverse and contain different features, different numbers of training examples, and different numbers of classes.
> - **SSL Methods** - To further enrich our study, we've added to VICReg and SimCLR, the clustering-based SSL method, SWaV, and Barlow Twins.
> - **Integration with Recent Architectures as Backbones** - Additional to RES-L-W of different widths and depths, we included Vision Transformer (VIT) [12] as an additional backbone.
>
> Thank you again for your thoughtful review. We made an effort to address your feedback including multiple experiments and paper edits, and we would greatly appreciate it if you would consider raising your score in light of our response.  Please let us know if you have additional questions we can address.
>
> ## References
> [1] V. Papyan, et al., "Prevalence of neural collapse during the terminal phase of deep learning training" 2020.
>
> [2] Z. Zhu, et al., "A Geometric Analysis of Neural Collapse with Unconstrained Features", 2021.
>
> [3] I. Ben-Shaul and S. Dekel, "Nearest Class-Center Simplification through Intermediate Layers", 2022
>
> [4] A. Bardes, et al., "VICReg: Variance-Invariance-Covariance Regularization for Self-Supervised Learning", 2022.
>
> [5] T. Chen, et al., "A Simple Framework for Contrastive Learning of Visual Representations", 2020.
>
> [6] J.-B. Grill, et al., "Bootstrap Your Own Latent a New Approach to Self-Supervised Learning", 2020.
>
> [7] M. Caron, et al., "Unsupervised Learning of Visual Features by Contrasting Cluster Assignments", 2020.
>
> [8] Y. Le et al., "Tiny ImageNet Visual Recognition Challenge," 2015.
>
> [9] S. Maji et al., "Fine-Grained Visual Classification of Aircraft", 2013.
>
> [10] Zhang, H. et al. "0/1 Deep Neural Networks via Block Coordinate Descent", 2022.
>
> [11] M.-E. Nilsback et al., "Automated Flower Classification over a Large Number of Classes", 2008.
>
> [12] A. Dosovitskiy et al., "An Image is Worth 16x16 Words: Transformers for Image Recognition at Scale", 2021.
>
> [13] M. Assran et al., "Self-Supervised Learning from Images with a Joint-Embedding Predictive Architecture", 2023

---

> > ### Comment · Reviewer_7nAJ · 2023-08-15
> > **RE: Rebuttal by Authors**
> >
> > I appreciate the author's detailed comments to address my concerns, especially the additional experiments using the backbones including ViT, the new datasets, and the SSL methods. I believe that these additional results improve the paper. I would like to maintain my recommendation as borderline accept.

---

> > > ### Author Response · Authors · 2023-08-15
> > >
> > > Thank you for your feedback, for engaging with us, and recommending acceptance of our submission!
> > >
> > > We are committed to the review process and to ensuring that our work meets the highest standard. Are there specific areas in our submission where further improvement would enable you to confidently recommend our paper for acceptance (i.e., an above-borderline acceptance recommendation)? We greatly appreciate your continued engagement!

---

> > > > ### Author Response · Authors · 2023-08-20
> > > > **Additional Experiments**
> > > >
> > > > Dear Reviewer 7nAJ,
> > > >
> > > > The discussion period ends in 24 hours.
> > > >
> > > > We have made a significant effort to address the questions and concerns raised in your review, including additional datasets (more than five different datasets), SSL methods (more than five methods), and backbone architectures (RES-L-W, Vision Transformers, and ResNets in different sizes). We would like to point out that we have even added analysis on the **ImageNet dataset for various SSL methods**, as mentioned in the general comment.
> > > >
> > > > We have made every effort to address your concerns in our revised manuscript through extensive experiments and clarifications. We kindly request that you consider these in your final assessment and consider raising your score in your final response.
> > > >
> > > > Please do not hesitate to let us know if there are additional points we can address.

---

### Author Rebuttal · Authors · 2023-08-09

Dear Reviewers,

Thank you for your thoughtful and detailed reviews of our work. We appreciate your time and the constructive feedback you have provided. We have carefully considered your comments and concerns and would like to address them in a unified response:

# 1. **Diversity of Architectures and Experiments**:
Several reviewers expressed concerns regarding the limited scope of our experimental studies, and some mistakenly believed that we only conducted experiments on CIFAR100 using the VICReg method. We recognize that we may need to be clearer in the paper about the diversity of our experiments, and we apologize for any confusion.

In the appendix of our original submission, we included additional results using CIFAR10 and FOOD101 datasets, as well as SimCLR as an SSL method. Additionally, we have included analysis using different random target functions, using both ResNet and ViT. We understand this information may not have been easily accessible, and we have taken steps to rectify this oversight in the revised manuscript.

In response to the reviewers' feedback, we have further expanded our analysis to include:
- **Datasets** - Additional to CIFAR100, CIFAR10, and FOOD101, we included results on Tiny-ImageNet [1], Aircrafts [2], Oxford Pets [3], and Oxford Flowers [4]. These datasets are much more diverse and contain different features, different numbers of training examples, and different numbers of classes.
- **SSL Methods** - To further enrich our study, we've added to VICReg and SimCLR, the clustering-based SSL method, SWaV, and Barlow Twins.
- **Integration with Recent Architectures as Backbones** - Additional to RES-L-W of different widths and depths, we included Vision Transformer (VIT) [5] as an additional backbone.

As you can see on the attached page, these new models, SSL methods, and architectures exhibit similar behavior to the current experiments. These additions, along with clarifying our existing results in the appendix, strengthen our paper by providing a more comprehensive and robust analysis.

# 2. **Originality and Quality**:
We addressed concerns raised about the originality of our paper by emphasizing our unique approach, specifically our analysis through the lenses of Neural Collapse, Randomness, and Information Compression. We also clarified the distinctions between our contributions and related works cited by the reviewers. Our expanded experiments and more detailed explanations have enhanced our paper's quality.

# 3. **Clarity and Presentation**:
In response to feedback on the clarity and flow of our manuscript, we have revised sections to provide clearer explanations and more explicit connections between our analysis and figures. We've ensured that our results are presented in an accessible manner and that our key insights are communicated effectively.

# 4. **Addressing Concerns and Questions**:
We addressed each reviewer's individual weaknesses and questions in detail. Please refer to the individual responses below for detailed explanations of how we addressed each concern.

**In conclusion**, we sincerely believe that the revisions and clarifications made in response to the reviewers' feedback have significantly strengthened our manuscript.

We have put forth a significant effort to address all your feedback, and your suggestions have improved our work.  We welcome further discussion, and we appreciate your valuable input. Again, Thank you for your time and insights that have greatly contributed to enhancing our work.

# References
[1] Y. Le et al., "Tiny ImageNet Visual Recognition Challenge", 2015.

[2] S. Maji et al., "Fine-Grained Visual Classification of Aircraft" , 2013.

[3] Zhang, H. et al. "0/1 Deep Neural Networks via Block Coordinate Descent", 2022.

[4] M.-E. Nilsback et al., "Automated Flower Classification over a Large Number of Classes", 2008

[5] A. Dosovitskiy et al., "An Image is Worth 16x16 Words: Transformers for Image Recognition at Scale", 2021.

---

### Author Response · Authors · 2023-08-19
**Additional Experiments on ImageNet**

Dear Reviewers,

We sincerely thank you for your constructive feedback and the time spent reviewing our manuscript. Your insights have been invaluable in identifying areas for further improvement and have contributed significantly to enhancing the quality of our manuscript.

In addition to our extensive experimental evaluation described above on many more datasets, which include different architectures and SSL methods, prompted by your feedback, we conducted several additional extensive experiments on the **ImageNet** dataset, which contains 1,281,167 images for training and 50,000 images for validation.

We have conducted further experiments using the ImageNet dataset with ResNet50 and several  SSL methods, including **VICReg, MoCoV2, SwAV, and BarlowTwins**. We are encouraged to observe trends similar to those identified in our previous analyses on other datasets. To share some specific numbers, the NCC  of the last layer for different SSL methods on the ImageNet dataset are as follows:

| **Init**         | **SwAV**       | **VICReg**     | **BarlowTwins** | **MoCoV2**     |
|------------------|----------------|----------------|-----------------|----------------|
| 0.03522386818    | 0.2700982024   | 0.4742010652   | 0.3506574567    | 0.4009445739   |

As evidenced by the results, the SSL methods contribute significantly to the clustering process, demonstrating strong clustering behavior as training progresses. These outcomes align with our broader observations and support our hypothesis and contributions.

Unfortunately, due to the open review system's limitations, we cannot share visual representations of these results at this point. We understand the impact figures can have in portraying results, and we are committed to including these in the final version of the paper, should it be accepted.

We are committed to integrating these new results into the camera-ready version of the paper and presenting them clearly through figures. This additional analysis significantly strengthens our work by demonstrating our findings for large datasets.

We specifically conducted the experiments you requested and hope you can consider these efforts in your evaluation. We kindly ask that you consider these contributions when reassessing our work, and we hope that they warrant an increase in our score.

Thank you for your constructive and insightful feedback. We appreciate your consideration of our paper and welcome further suggestions.

---

### Decision · Program_Chairs · 2023-09-21

**Decision:**

Accept (poster)

**Comment:**

The paper provides an in-depth analysis of self-supervisedly learned visual representations from the perspective of latent semantic clustering. While the initial paper was still lacking larger-scale (imagenet) results, this was sufficiently addressed in the rebuttal to the satisfaction of the reviewers. As a result, the reviewers find this paper provides an important study with qualitative and quantitative results that is clearly written